# A naturally selected αβ T cell receptor binds HLA-DQ2 molecules without co-contacting the presented peptide

Jia Jia Lim [1], Claerwen M. Jones [1], Tiing Jen Loh [1], Hien Thy Dao[1], Mai T. Tran[1], Jason A. Tye-Din [2], Nicole L. La Gruta [1,4] ✉ & Jamie Rossjohn [1,3,4] ✉

αβ T cell receptors (TCR) co-recognise peptide (p) antigens that are presented by major histocompatibility complex (MHC) molecules. While marked variations in TCR-p-MHC docking topologies have been observed from structural studies, the co-recognition paradigm has held fast. Using HLA-DQ2.5-peptide tetramers, here we identify a TRAV12-1⁺-TRBV5-1⁺ G9 TCR from human peripheral blood that binds HLA-DQ2.5 in a peptide-agnostic manner. The crystal structures of TCR-HLA-DQ2.5-peptide complexes show that the G9 TCR binds HLA-DQ2.5 in a reversed docking topology without contacting the peptide, with the TCR contacting the β1 region of HLA-DQ2.5 and distal from the peptide antigen binding cleft. High-throughput screening of HLA class I and II molecules finds the G9 TCR to be pan-HLA-DQ2 reactive, with leucine-55 of HLA-DQ2.5 being a key determinant underpinning G9 TCR specificity excluding other HLA-II allomorphs. Consistent with the functional assays, the interactions of the G9 TCR and HLA-DQ2.5 precludes CD4 binding, thereby impeding T cell activation. Collectively, we describe a naturally selected αβTCR from human peripheral blood that deviates from the TCR-p-MHC co-recognition paradigm.

Major Histocompatibility Complex (MHC) molecules present a wide array of peptide antigens on the surface of antigen presenting cells[1]. αβ T cell receptors (TCRs), expressed on the surface of T cells, become activated upon recognition of the antigenic peptide presented by MHC molecules. For CD4⁺ T cells, effective T cell signalling requires the CD4 co-receptor that acts as a modulator by recruiting Src tyrosine kinase p65lck (Lck) to the TCR-peptide (p)-MHC complex, thereby promoting phosphorylation of the immunoreceptor tyrosine activation motifs (ITAMs) in the TCR-CD3 complex[2]. Central to this TCR-pMHC recognition event, the TCRs specifically and simultaneously co-recognise peptide antigens presented by the MHC molecules. Indeed, T cells within a given individual generally only recognise peptides presented by MHC molecules expressed by that individual[3].

Since 1996, extensive structural and functional studies in the TCR-pMHC axis have detailed the molecular basis of the diverse and unique TCR ligand specificity and sensitivity in mediating T cell immunity[1,4,5]. Here, all αβTCRs sit atop of the MHC antigen-binding cleft in an 'end-to-end-manner', co-contacting the peptide antigen and MHC molecules. For the vast majority of TCR-pMHC structures solved, the TCR binds peptide-MHC with a canonical polarity, namely the TCR α-chain orientated on top of the β/α2-chain, and the TCR β-chain positioned over the α/α1-chain in MHC class II or class I molecules, respectively[4,6,7]. Recently, however, reversed TCR-pMHC polarity modes have been observed, whereby the TCR α- and β- chains were oriented 180° with respect to the MHC class II and I molecules[8,9]. This indicated that the canonical TCR docking over pMHC is essential for T cell signalling,

¹Infection and Immunity Program and Department of Biochemistry and Molecular Biology, Biomedicine Discovery Institute, Monash University, Clayton, VIC, Australia. ²Immunology Division, The Walter and Eliza Hall Institute, Parkville, VIC, Australia. ³Institute of Infection and Immunity, Cardiff University School of Medicine, Heath Park, Cardiff, UK. ⁴These authors contributed equally: Nicole L. La Gruta, Jamie Rossjohn. ✉e-mail: nicole.la.gruta@monash.edu; jamie.rossjohn@monash.edu

whereas the reverse docking TCRs either elicit apparent inherent MHC-II autoreactivity[8] or impeded MHC-I restricted T cell activation due to mislocalisation of Lck relative to the CD3 complex[9].

In this study, we isolate T cells restricted to HLA-DQ2.5$^{glia-\omega1}$ (PFPQPEQPF) or HLA-DQ2.5$^{glia-\omega2}$ (PQPEQPFPW) from the peripheral blood of a gluten challenged coeliac disease donor and unexpectedly find a *TRAV12-1⁺-TRBV5-1⁺* CD4⁺ T cell that elicited peptide-independent characteristics using in-vitro cell tetramer staining and surface plasmon resonance (SPR) affinity analysis. The crystal structures of the TRAV12-1⁺-TRBV5-1⁺G9 TCR complexed with HLA-DQ2.5$^{glia-\omega1}$, HLA-DQ2.5$^{CLIP}$ (ATPLLMQALPMGA), and HLA-DQ2.2$^{glutL1}$(QPPASEQEQPVLP), uncover a reproducible and unanticipated binding mode of TCR-pMHC, whereby the G9 TCR binds at the β1 of HLA-DQ2, away from the peptide binding cleft. Accordingly, we provide insight into the specificity determinants underlying G9 TCR-HLA-DQ2 recognition and confirmed that the G9 TCR is pan-HLA-DQ2 specific, but did not react with other HLAs.

## Results

### Discovery of a peptide independent CD4⁺ T cell restricted to HLA-DQ2.5

Using tetramer based magnetic enrichment[10,11] and single cell index sorting approaches, we isolated CD4⁺ T cells binding to HLA-DQ2.5 (*DQA1*05:01/DQB1*02:01*) tetramers presenting immunodominant dietary gluten epitopes, glia-ω1 and glia-ω2 (PFPQPEQPF and PQPEQPFPW, respectively) from the peripheral blood of a HLA-DQ2.5 homozygous coeliac disease donor six days after wheat challenge (Fig. 1a and Supplementary Fig. 1a). Paired TCRαβ usage was determined by multiplex PCR and sequencing. While there appeared to be some preferential usage of TCRs using the *TRAV19* gene element in cells binding the HLA-DQ2.5$^{glia-\omega2}$ tetramer, TCRαβ usage of both HLA-DQ2.5$^{glia-\omega1}$ and HLA-DQ2.5$^{glia-\omega2}$ repertoires appeared to be diverse, with few clones extensively expanded (Fig. 1a). To confirm specificity, we transiently expressed a selection of these paired αβTCRs in 293 T cells and stained with individual HLA-DQ2.5 tetramers presenting gluten epitope glia-α1 (PFPQPELPY), glia-α2 (PQPELPYPQ), glia-ω1 (PFPQPEQPF), or glia-ω2 (PQPEQPFPW), as well as HLA-DQ8$^{glia-\alpha1}$ (GEGSFQPSQENP) as a control (Fig. 1b, Supplementary Fig. 1b, 2a & 2b, and Supplementary Table 1). Intriguingly, a TRAV12-1⁺-TRBV5-1⁺ G9 TCR bound to HLA-DQ2.5 tetramers presenting any of the four individual gluten epitopes with a very similar staining pattern, but did not bind to HLA-DQ8$^{glia-\alpha1}$ tetramer (Fig. 1b). These data suggested that the binding was HLA-DQ2.5 specific and independent of peptide. This potential peptide-independent HLA-DQ2.5 specificity was not observed for any of the other tested TCRs, even those with similar tetramer binding patterns as the G9 TCR (Fig. 1a). Instead, these TCRs were specific for DQ2.5$^{glia-\omega1}$ or DQ2.5$^{glia-\omega2}$, or in one case, namely clone B01, cross-reactive for the highly homologous DQ2.5$^{glia-\omega1}$ (PFPQPEQPF) and DQ2.5$^{glia-\alpha1}$ (PFPQPELPY) epitopes (Supplementary Fig. 2a).

To determine the role of peptide antigens in recognition of G9 TCR and HLA-DQ2, we expressed and purified the soluble G9 TCR and determined steady-state binding affinities ($K_D$) of the TCR for their respective pHLA via surface plasmon resonance (SPR). The G9 TCR revealed a comparable affinity for HLA-DQ2.5 (*DQA1*05:01/ DQB1*02:01*) bound to glia-α1, −α2, −ω1, −ω2, or CLIP, with a $K_D$ of ~7–14 μM (Fig. 1c). Moreover, G9 TCR cross-reacted with HLA-DQ2.2$^{glutL1}$ (*DQA1*02:01/DQB1*02:02*) with a similar $K_D$ of 12.4 μM (Fig. 1c), consistent with either extensive TCR cross-reactivity across a series of pHLA-II complexes or a peptide independent binding mode.

### Structural basis of G9 TCR-peptide-HLA-DQ2 interaction

To understand the mechanism underpinning recognition of TRAV12-1⁺-TRBV5-1⁺ G9 TCR, we solved the crystal structures of G9 TCR in complex with HLA-DQ2.5$^{glia-\omega1}$, HLA-DQ2.5$^{CLIP}$, and DQ2.2$^{glutL1}$ to 2.20 Å, 2.45 Å, and 2.20 Å resolution, respectively (Fig. 2a, Supplementary Fig. 3, and Supplementary Table 2). Notably, G9 TCR docked atop of

HLA-DQ2 β-chain, away from peptide-antigen binding cleft, consistent with the cell staining and affinity analyses (Fig. 2a). Alignment of three complexes on Cα backbone of HLA-DQ2.5$^{glia-\omega1}$ complex revealed a highly conserved pattern of G9 TCR CDR loops and HLA-DQ2 recognition, with a root mean square deviation (r.m.s.d) value of 0.13 Å and 0.35 Å for HLA-DQ2.5$^{CLIP}$ complex, and DQ2.2$^{glutL1}$ complex, respectively (Supplementary Fig. 4a). The HLA-DQ2 peptide binding cleft was rigid with very limited deviation in the helix region of DQ2 α- and β-chain (Supplementary Fig. 4b). Despite relatively low sequence identity of glia-ω1 and CLIP (27%) or glutL1 (36%) peptides, the binding register of the peptide bound to HLA-DQ2 was conserved (Supplementary Fig. 4b, and Supplementary Table 1). Moreover, the G9 TCR-HLA-DQ2 interfaces were comparable in all three complexes, with the total buried surface area (BSA) for the G9 TCR being -1400 A$^2$, falling within the typical range of relative BSA values observed for TCR-pMHC II structures[1] (Fig. 2b, Supplementary Fig. 4c, and Supplementary Table 3). In the G9 TCR-HLA-DQ2.5$^{glia-\omega1}$ complex, the Vα-chain of G9 TCR dominated the HLA-DQ2 interface, comprising 59% of total BSA, where the CDR1α, CDR3α, and framework-α (FWα) regions of the G9 TCR contributed 31%, 22%, and 6% to the BSA, respectively (Fig. 2b). Whereas the Vβ-chain of G9 TCR contributed 41% to the BSA with the CDR2β (14%), CDR3β (18%) and FWβ (9%) of G9 TCR interacting with the HLA-DQ2 β-chain (Fig. 2b and Supplementary Table 3). No contacts were made with the HLA-DQ2 α-chain and glia−ω1 peptide, with the closest distance between CDR loops and peptide being -17 Å (Fig. 2c).

The CDR1α and FWα positioned over the N-terminal region of the α−helix of the HLA-DQ2.5 β−chain while CDR3α, CDR3β, CDR1β, and FWβ oriented towards the β−sheet region of HLA-DQ2.5 β−chain (Fig. 2c). Leu$^{52}$β, Leu$^{55}$β, and Glu$^{59}$β residues located adjacent to the peptide antigen binding cleft of the HLA-DQ2.5 β−chain made extensive contacts with CDR1α loop (Ser$^{28}$α, Ala$^{29}$α, Ser$^{36}$α, and Gln$^{37}$α) and FWα (Arg$^{82}$α) (Fig. 2d, e, and Supplementary Table 4). Moreover, Gln$^{37}$α residue in the CDR1α loop also contacted distal region of HLA-DQ2.5 β−sheet by forming multiple H-bonds and VdW interactions with Phe$^{47}$β, Arg$^{48}$β, Ala$^{49}$β, Leu$^{55}$β, Glu$^{59}$β, and Gln$^{62}$β residues (Fig. 2d and Supplementary Table 4). The CDR3α loop sat centrally on the β−sheet region of HLA-DQ2.5, where Met$^{107}$α, Phe$^{109}$α, and Gln$^{110}$α formed multiple polar and hydrophobic contacts with Glu$^{46}$β, Arg$^{48}$β, Ala$^{49}$β, Val$^{50}$β, Arg$^{39}$β of HLA-DQ2.5 β−chain (Fig. 2e and Supplementary Table 4). The CDR2β of G9 TCR was also stabilized by the β−sheet region of HLA-DQ2.5 β−chain, whereby the Glu$^{63}$β, Ser$^{58}$β and Phe$^{57}$β formed multiple contacts with Arg$^{23}$β and Asp$^{43}$β of HLA-DQ2.5 β−chain via salt bridges, H-bonds, and VdW interactions (Fig. 2f and Supplementary Table 4). Moreover, Arg$^{66}$β of the FWβ made key interactions, namely, 2 salt bridges with Asp$^{41}$β and Asp$^{43}$β, and VdW interactions with Val$^{44}$β, and Arg$^{25}$β, thus stabilizing the loop of HLA-DQ2.5 β−sheet (Fig. 2f and Supplementary Table 4). The CDR3β loop of the G9 TCR spanned the α-helix and β−sheet of the HLA-DQ2.5 β−chain (Fig. 2g). The polar-mediated contacts between Arg$^{109}$β, Ala$^{110}$β and Glu$^{111}$β of CDR3β with Arg$^{48}$β, Gly$^{45}$β, Val$^{44}$β, Asp$^{43}$β, and Arg$^{72}$β of HLA-DQ2.5 further increased the HLA interactions (Fig. 2g and Supplementary Table 4). Detailed interactions of G9 TCR and HLA-DQ2.5$^{CLIP}$ or DQ2.2$^{glutL1}$, respectively, are highly conserved, with very subtle deviations at HLA-DQ2 (Arg$^{23}$ and Asn$^{62}$) and G9 TCR (Gln$^{110}$) (Supplementary Fig. 4d−g and Supplementary Table 4−6). Collectively, the CDR1α, CDR3α, CDR2β, and CDR3β loops of the G9 TCR enabled specific contacts with the HLA-DQ2.5β−chain, while no contacts were made with the HLA-DQ2.5α−chain.

### Energetic basis of TRAV12-1⁺-TRBV5-1⁺ G9 TCR and HLA-DQ2

To characterise the energetic basis for the *TRAV12-1⁺-TRBV5-1⁺* usage of G9 TCR, we generated fifteen single site alanine-scanning mutations on the TCR based on the crystal structures and analysed their impact in binding HLA-DQ2.5 presenting CLIP or glia-ω1 peptide or HLA-DQ2.2

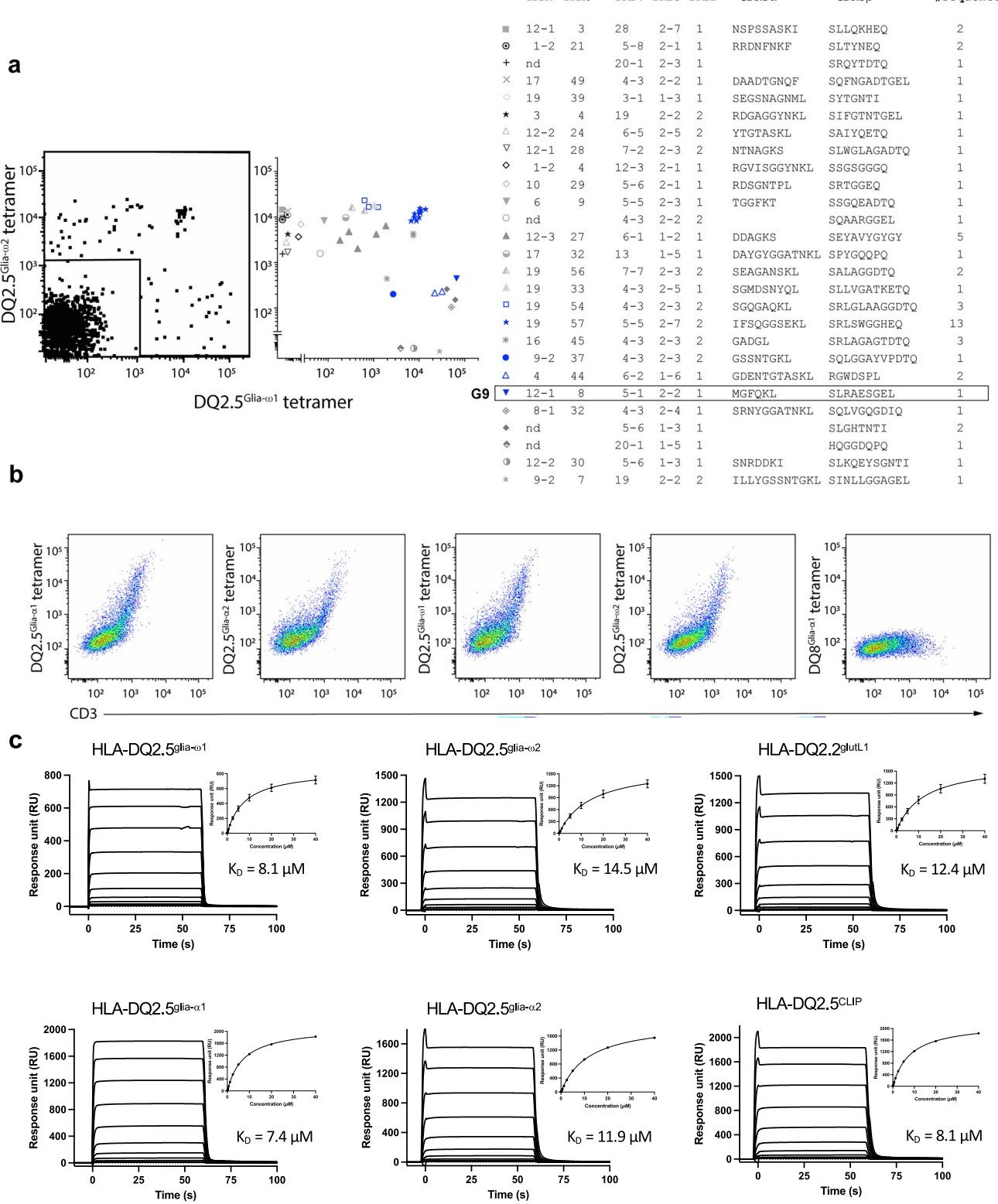

**Fig. 1 | Identification of a peptide-independent CD4⁺ T cell restricted to HLA-DQ2.5. a** HLA-DQ2.5$^{glia-ω1}$ and DQ2.5$^{glia-ω2}$ tetramer co-staining on CD4⁺ T cells post tetramer-based magnetic enrichment of PBMC of wheat challenged coeliac disease donor #0648. Right plot represents expansions of individual clones, determined via index sorting, with TCR gene segment usage, CDR3 sequence and frequency shown for each clone. Box indicates cell designated G9 TCR. **b** HEK 293 T cells transiently co-transfected with G9 TCR and CD3γδεζ were stained with individual HLA-DQ2.5 tetramers presenting glia-α1, glia-α2, glia-ω1, or glia-ω2, respectively, or control HLA-DQ8$^{glia-α1}$ tetramer. **c** Affinity measurement of G9 TCR against HLA-DQ2.5$^{glia-α1/glia-α2/glia-ω1/glia-ω2/CLIP}$ and HLA-DQ2.2$^{glut-L1}$ interactions. HLA-DQ8$^{glia-α1}$ was immobilised in the reference flow cell to control non-specific binding. For $K_D$ determination, all data were derived from two independent experiments in duplicate and curve fits using a 1:1 binding model. For each concentration, the points represent the mean and the error bars correspond to ± s.e.m.

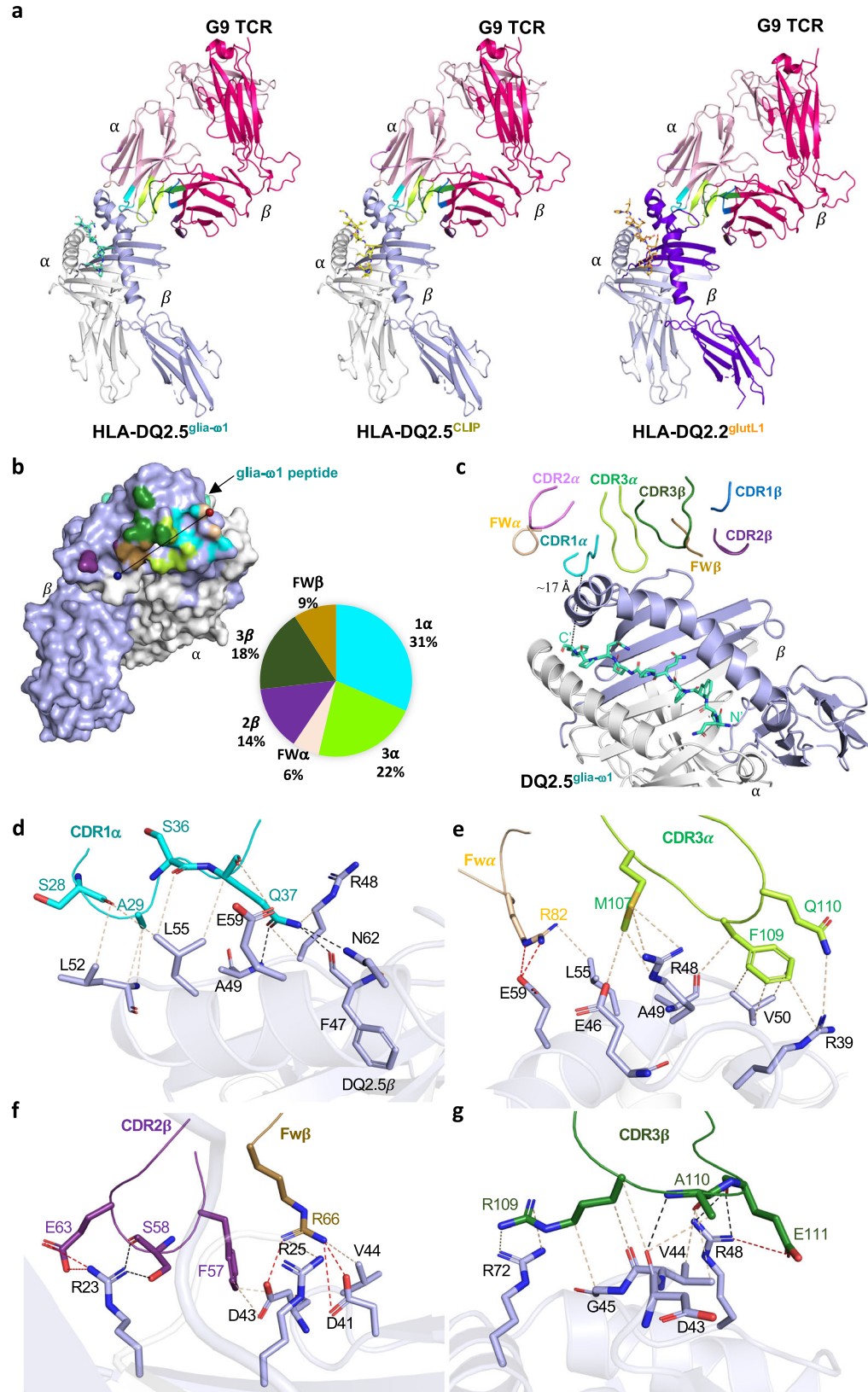

presenting glutL1 peptide using SPR. Twelve residues of CDR loops that form multiple contacts (H-bonds, VdW and/or salt bridge) with HLA-DQ2, and three residues located at the crystallographic packing region or at the interface with no contact with HLA-DQ2 were selected. The pattern of effects of the mutants was consistent across all three peptide epitopes, in line with the peptide-independent mode of

recognition (Fig. 3a and Supplementary Fig. 5). The impact of TCR mutants on HLA-DQ2 interactions was categorised into four groups: no effect (≤ 2-fold reduced affinity compared to wildtype TCR), moderate effect (3-5-fold reduced affinity), substantial effect (5-10-fold reduced affinity) and critical effect (>10-fold reduced affinity). As expected, mutation of Gln$^{48\beta}$, Arg$^{75\beta}$ (FWβ), and Ser$^{112\beta}$ (CDR3β), which served as

**Fig. 2 | TRAV12-1⁺/TRBV5-1⁺ G9 TCR recognition of HLA-DQ2.5/DQ2.2. a** Overall structure representation of G9 TCR in complex with HLA-DQ2.5^glia-ω1, HLA-DQ2.5^CLIP and HLA-DQ2.2^glutL1 molecules. The HLA-DQ2.5 α- and β-chains are coloured in white and light blue, respectively, whereas the HLA-DQ2.2 α- and β-chains are coloured in grey and purple, respectively. The glia-ω1, clip, and glutL1 peptides are coloured in green, yellow, and orange sticks, respectively. The CDR loops 1α, 2α, and 3α are highlighted in cyan, violet, and light green colour, whereas 1β, 2β, 3β are coloured in blue, purple, and dark green, respectively. The FWα residues are coloured in beige and FWβ residues are colour in sand. **b** Surface representation of G9 TCR footprint on HLA-DQ2.5 β-chain. TCR footprint colours are in accordance with the nearest TCR contact residue. The Vα and Vβ centre of mass position are shown in red and blue spheres, respectively, and connected via a black line. Pie chart represents the relative contribution of each CDR loop and FW residues of the G9 TCR to the interface with HLA-DQ2.5^glia-ω1. **c** Overall G9 TCR CDR loops docking on HLA-DQ2.5^glia-ω1 and detailed interactions of G9 TCR between (**d**) CDR1α, (**e**) FWα and CDR3α, (**f**) CDR2β and FWβ, (**g**) CDR3β with HLA-DQ2.5^glia-ω1 are shown. The H-bonds, VdW, and salt bridges were displayed as black, light beige, and red dash lines, respectively. All amino acids are indicated in single-letter abbreviations.

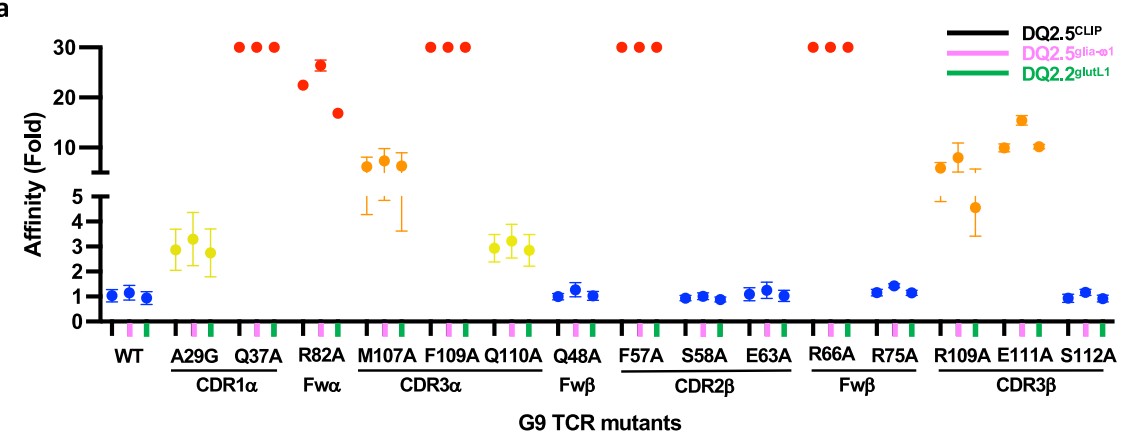

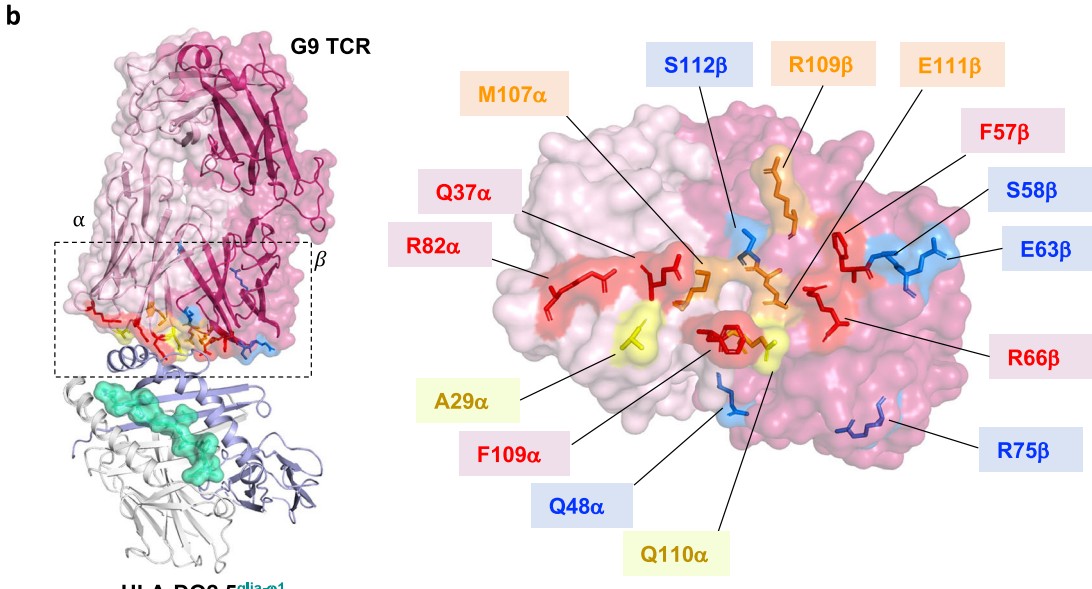

**Fig. 3 | Energetic basis of TRAV12-1⁺/TRBV5-1⁺ G9 TCR recognition of HLA-DQ2. a** Effect of G9 TCR point mutations at the HLA-DQ2 interface. The Y-axis represents the fold of affinity of mutant TCR in comparison to the wild-type (WT) TCR. The X-axis represents the position of G9 TCR mutants at CDR loops and framework region. The affinity of G9 TCR mutants against HLA-DQ2.5^glia-ω1 (denoted as pink line), HLA-DQ2.5^CLIP (denoted as black line) and HLA-DQ2.2^glutL1 (denoted as green line) were performed via SPR ($n = 3$) and the error bars correspond to ± s.e.m. The impact of each mutation was classified as negligible (≤ 2-fold affinity decrease, blue), moderate (3-5-fold affinity decrease, yellow), substantial (5-10-fold affinity reduction, orange), or critical (>10-fold affinity decrease or no binding, red) shown on graph and on surface of HLA-DQ2. **b** Left: overall representation of the G9 TCR-HLA-DQ2.5 interface, right: the footprint of energetically important G9 TCR contact residues were shown as sticks and coloured according to the changed $K_D$ in fold as **a**. All amino acids are indicated in single-letter abbreviations.

non-HLA contacting control residues had no impact on the TCR-HLA interaction (Fig. 3a). Similarly, mutations at Ser^58β and Glu^63β (CDR2β), had no effect on the affinity of the interaction, indicating the non-essential role of these two residues in mediating contacts with HLA-DQ2. Moreover, the AlaGly mutation (CDR1α) and Gln^110α-Ala mutation (CDR3α) showed a moderate impact on HLA-DQ2 recognition, with a 3-5-fold reduced affinity compared to wildtype G9 TCR (Fig. 3a, b).

In contrast, the CDR1α−Gln^37, FWα-Arg^82 and CDR3α-Phe^109 residues that formed extensive contacts with HLA-DQ2β-chain had

critical effects upon alanine substitution (Figs. 2d, e, 3a, b). The CDR3α-Met^107 Ala mutation also had a substantial effect on binding to HLA-DQ2, with a 5-10-fold reduction in affinity (Fig. 3a, 3b). For the β-chain of the G9 TCR, alanine substitution of Phe^57β (CDR2β), Arg^66β (FWβ), Arg^109β and Glu^111β (CDR3β) showed substantial impact on HLA-DQ2 recognition (>5-fold reduced affinity) (Fig. 3a, b). Accordingly, eight residues located in the germline encoded and non-germline encoded regions of the G9 TCR - four residues each from the TCR α and β-chains - formed a core hotspot at the HLA-DQ2 β-chain,

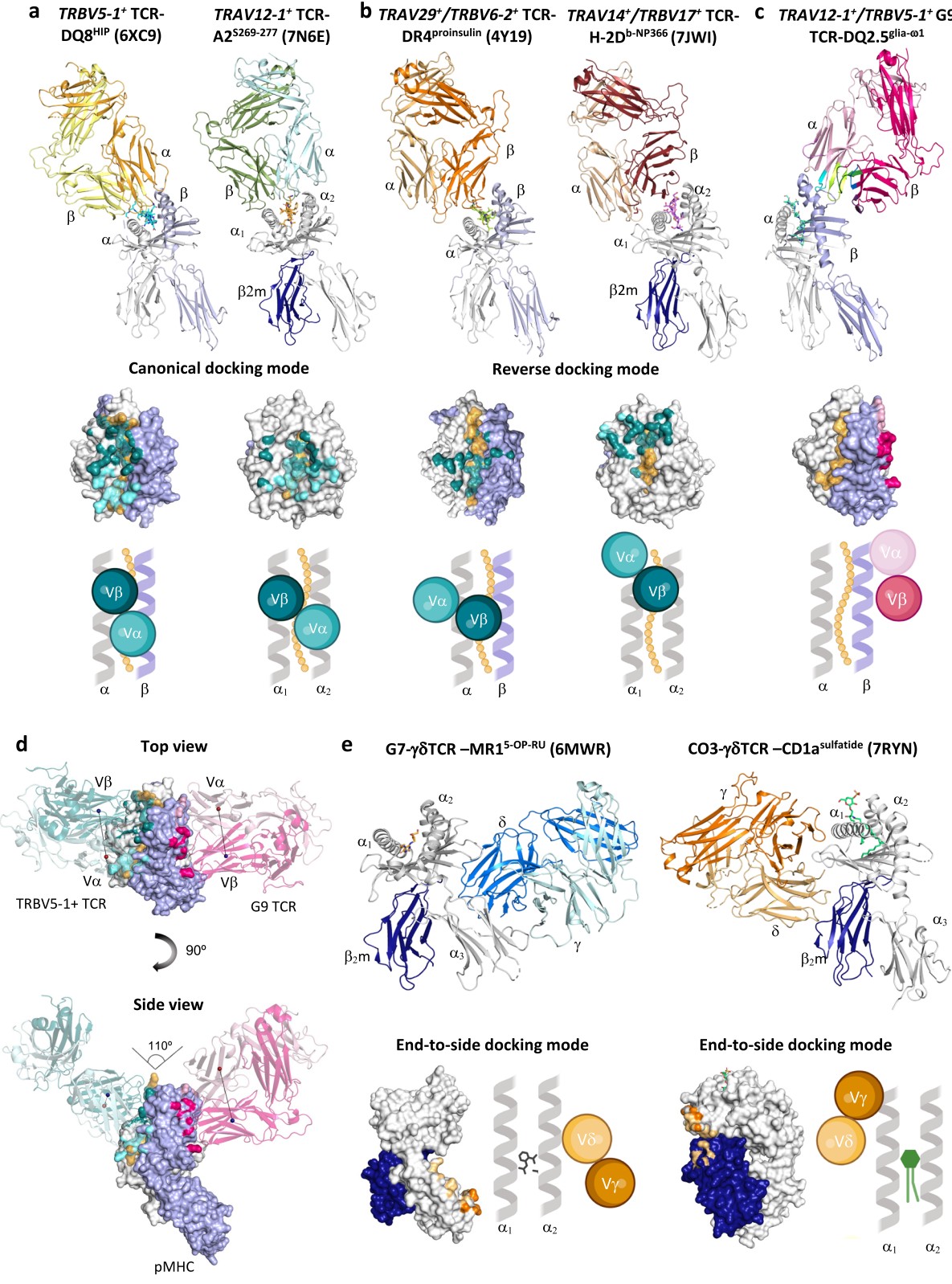

highlighting the importance of *TRAV12-1⁺-TRBV5-1⁺* pairing for G9 TCR-HLA-DQ2 recognition (Fig. 3b).

## Atypical binding mode of G9 TCR-HLA-DQ2 recognition

After nearly three decades of αβTCR-pMHC structural studies, the mode of TCR recognition has been categorised into canonical and reversed polarity docking, where the canonical docking binding mode

predominates[4]. For canonical docking, the αβTCR sits atop of pMHC, with the Vα chain located on top of β/α2−chain, and Vβ chain located on top of α/α1−chain in MHC class II or class I molecules, as exemplified in TRBV5-1⁺ TCR-HLA-DQ8^HIP (PDB ID: 6XC9)[12] and TRAV12-1⁺ TCR-HLA-A2^S269-277 (PDB ID: 7N6E)[13] complexes, respectively (Fig. 4a).

By contrast, in reversed polarity docking, for which there are presently only two examples in the literature, the Vα and Vβ of the TCR

**Fig. 4 | A new binding mode of G9 TCR and HLA-DQ2. a** Schematic representation of canonical docking mode of TCR on MHC class II (TRBV5-1⁺ TCR-HLA-DQ8^HIP; PDB: 6XC9) and class I molecule (TRAV12-1⁺ TCR-HLA-A2^S269-277; PDB: 7N6E). **b** Overall structure, surface representation, and schematic representation of reverse docking mode of TCR on MHC class II presenting peptide antigens (TRAV29⁺/TRBV6-2⁺ TCR-HLA-DR4^Proinsulin; PDB: 4Y19) and class I molecule (TRAV14⁺/TRBV17⁺ TCR-HLA-H-2D^b-NP366; PDB: 7JWI). For TCR footprint and schematic docking mode view in (**a**) and (**b**), the TCR is coloured in accordance with the in TCR α- (light teal) and β-chains (dark teal) contact residues. The MHC class II α- and β-chains are coloured in white and light blue, respectively, whereas MHC class I α-chain and β2m are coloured in white and dark blue, respectively. The peptide is coloured in orange. **c** Schematic overview of TRAV12-1⁺TRBV5-1⁺ G9 TCR-HLA-DQ2^glia-ω1. G9 TCRα- and β-chains footprint is coloured light pink and hot pink, respectively. The HLA-DQ2.5 α- and β- chains are coloured in white and light blue, respectively. **d** Top and side view of superposed structure of canonical TCR docking in (**a**) and G9 TCR docking in **c**. Spheres represent the center of mass of Vα (red) and Vβ (blue) of TCR. Canonical TCR (TRBV5-1⁺ TCR-HLA-DQ8^HIP; PDB: 6XC9) is coloured with the in TCR α- (light teal) and β-chains (dark teal). G9 TCRα- and β-chains is coloured light pink and hot pink, respectively. The HLA-DQ2.5 α- and β-chains are coloured in white and light blue, respectively. The peptide is coloured in orange. **e** Schematic representation of end-to-side docking mode of γδTCR on MHC class I-like molecule presenting small molecule (left) and lipid molecule (right). Left: G7- γδTCR-MR1^5-OP-RU; PDB: 6MWR; Right: CO3-γδTCR-CD1a^sulfatide; PDB: 7RYN. For TCR footprint and schematic docking mode, the TCRγ- and δ-chains are coloured in orange and beige, respectively. MHC class I α-chain and β2m chain are coloured in white and dark blue, respectively.

are oriented 180° with respect to the MHC molecule, where the Vα chain docks on top of α/α1−chain, and Vβ chain docks on top of β/α2−chain in MHC class II (TRAV29⁺/TRBV6-2⁺ TCR-HLA-DR4^proinsulin; PDB ID: 4Y19)[8] or class I molecules (TRAV14⁺/TRBV17⁺ B17.R2-TCR-HLA-H-2D^b-NP366; PDB ID: 7JWI)[9], respectively (Fig. 4b). In both cases, the αβTCRs bound across the pMHC with 'end-to-end' orientation, enabling interactions with both peptide and MHC (Fig. 4a, b). In this study, we show a new αβTCR-pMHC binding mode, whereby the Vα and Vβ of G9 TCR are oriented in the reversed polarity mode (-180°) and docked at an extreme angle (-110°) at the β1 region of HLA-DQ2^glia-ω1, away from the peptide binding cleft (Fig. 4c, d). This recognition is distinct from all currently available αβTCR-pMHC structures determined (Fig. 4a−c). The atypical αβG9−TCR-MHC recognition was more akin to the recently described γδTCR-MHC class I-like complex structures. Namely, γδTCR-MHC class I-like complexes demonstrated a 'end-to-side' recognition mode where the γδTCR was docked to the side of α2/α3 in MR1^5-OP-RU molecule (PDB ID: 6MWR)[14] or to the side of α1/β2m in CD1a^sulfatide molecule (PDB ID: 7RYN)[15] (Fig. 4e). Accordingly, the G9 TCR-HLA-DQ2 complex exhibited an unexpected 'end-to-side' TCR-pMHC docking topology in a reversed polarity.

### Glu⁴⁶ and Leu⁵⁵ residues in HLA-DQ2β determines G9 TCR specificity

To further characterise the specificity of the G9 TCR, we performed a high throughput Luminex screening across 180 individual HLA-class I and class II molecules that present a heterogeneous array of self-peptides. PE-conjugated G9 TCR tetramers were incubated with microbeads coated with individual HLA-class I and class II molecules, and subsequent mean fluorescence intensity (MFI) of each HLA allotype was analysed (Fig. 5a and Supplementary Fig. 6). As expected, the G9 TCR displayed specific binding to all HLA-DQ2 (*HLA-DQB1*02*) molecules, but displayed no binding to other HLA-class II (HLA-DQ, -DP, -DR) and HLA-class I (HLA-A, -B, and C) molecules, affirming the G9-HLA-DQ2 recognition is *DQB1*02* specific (Fig. 5a and Supplementary Fig. 6).

Multiple sequence alignment of a panel of HLA-DQ allomorphs including DQ2 (DQB1*02:01, *02:02), DQ4 (DQB1*04:01, *04:02), DQ5 (DQB1*05:01, *05:02), DQ6 (DQB1*06:01, *0602), DQ7 (DQB1*03:01) and DQ8 (DQB1*03:01, *03:02) showed that 14 out of 17 interacting residues that participated in G9 TCR-HLA-DQ2 recognition were conserved throughout the HLA-DQ allomorphs (Fig. 5b). Notably, three residues, namely Glu⁴⁶, Leu⁵² and Leu⁵⁵ were conserved in HLA-DQ2 (DQB1*02:01, *02:02), but were substituted with a Val⁴⁶, Pro⁵², and Pro⁵⁵/Arg⁵⁵ in other HLA-DQ allomorphs (DQB1*03, *04, *05, *06), suggesting the residues were likely to play a critical role in the selective G9 TCR binding to HLA-DQ2 molecules (Fig. 5b).

To examine this further, we generated a panel of eleven alanine mutations on the HLA-DQ2.5 molecule based on key interactions from our structural data, which includes Arg²⁹β, Arg³⁹β, Asp⁴³β, Val⁴⁴β, Glu⁴⁶β, Arg⁴⁸β, Leu⁵²β, Leu⁵⁵β, Glu⁵⁹β, Asp⁶⁶β, and Arg⁷⁷β, and evaluated the impact of G9 TCR-HLA-DQ2 affinity using SPR approach. Asp⁶⁶β

and Arg⁷⁷β acted as control, which had no impact to the affinity of G9 TCR-HLA-DQ2 complex (Fig. 6a, b, and Supplementary Fig. 7). Consistent with G9 TCR mutagenesis data, mutation of six out of seven key residues, namely Arg³⁹β, Asp⁴³β, Val⁴⁴β, Glu⁴⁶β, Arg⁴⁸β, and Leu⁵⁵β and Glu⁵⁹β had a critical effect on the interaction with G9 TCR (Fig. 6a, b, Supplementary Figs. 7 and 8), whereas Leu⁵²β and Arg²⁹β had limited impact on the affinity of G9 TCR (Fig. 6a, b, Supplementary Figs. 7 and 8). Structural superposition of HLA-DQ2.5 and HLA-DQ8 at the G9 TCR-HLA-DQ2.5 interface highlighted the importance of Glu⁴⁶ and Leu⁵⁵ in interacting with CDR1α and CDR3α loops of G9 TCR, thus defining the specificity towards HLA-DQ2 molecule only (Fig. 6b and Supplementary Fig. 8). Accordingly, we define the key residues (Glu⁴⁶ and Leu⁵⁵) of HLA-DQ2 in playing an essential role in G9 TCR binding.

### HLA-DQ2 binding by G9 TCR does not activate T cell signalling

Given the unusual binding mode of the G9 TCR-HLA-DQ2 complex, we investigated whether the peptide independent G9 TCR-HLA-DQ engagement led to T cell signal transduction. We first generated a stably expressing G9 TCR and a positive control G2-TCR (HLA-DQ2.5^glia-α1/ω1 cross-reactive clone) in the SKW3 T cell line, and subsequently determined upregulation of the early T cell activation marker CD69 as the readout for activation. Unlike the control G2-TCR line, the G9 TCR did not activate CD69 T cell signalling pathway upon co-culture with HLA-DQ2.5⁺ RAJI B Lymphoma cells and glia-peptides (Fig. 7a), despite the stably expressed SKW3 G9 TCR showing high CD3 expression and being well-stained with individual HLA-DQ2.5 tetramers presenting glia-peptides (Supplementary Fig. 1d and 9a).

To understand whether the lack of T cell signal transduction was because the presence of CD4 perturbed the coreceptor-associated Lck-CD3 signalling, we then created a stably expressing CD4 negative Jurkat clone using CRISPR-Cas9. Here, the CD4⁻ and CD4⁺ Jurkat cells were transduced with G9 TCR, and bulk G9 TCR cell lines were tested in a peptide stimulation assay with HLA-DQ2.5⁺ RAJI cells coated with glia-α1 or glia-ω1 peptide. CD4⁺ Jurkat cells expressing the G2-TCR were again included as a control. The G9 TCR was not stimulated by HLA-DQ2.5, even when CD4 was absent. CD4 negative and positive Jurkat cells expressing the G9 TCR did not show CD69 upregulation to DQ2.5-glia-α1 or -glia-ω1 despite both cell lines responding to anti-CD3 stimulation, whereas the control G2-TCR cell line revealed high CD69 upregulation in response to peptide stimulation (Fig. 7b). Jurkat cells expressing G9 TCR bound DQ2.5^glia-ω1 tetramer better than the G2-TCR-expressing cells, indicating that lack of G9 TCR stimulation in the assay is not affinity related (Supplementary Fig. 1c and 9b).

Next, we superposed the G9 TCR-HLA-DQ2 complex with the previously published αβTCR-pMHC-CD4 (MS2-3C8-TCR-HLA-DR4^MBP-CD4; PDB ID:3T0E)[16] complex. In the canonical docking mode, the CD4 complex (D1-D2-D2-D4 domains) bound to the constant domain of HLA, forming an arch-like architecture. Whereas in the superposed model structure of G9 TCR-HLA-CD4, the G9 TCR docking interfered with the CD4 complex architecture at D3 and D4 domains (Fig. 7c). The

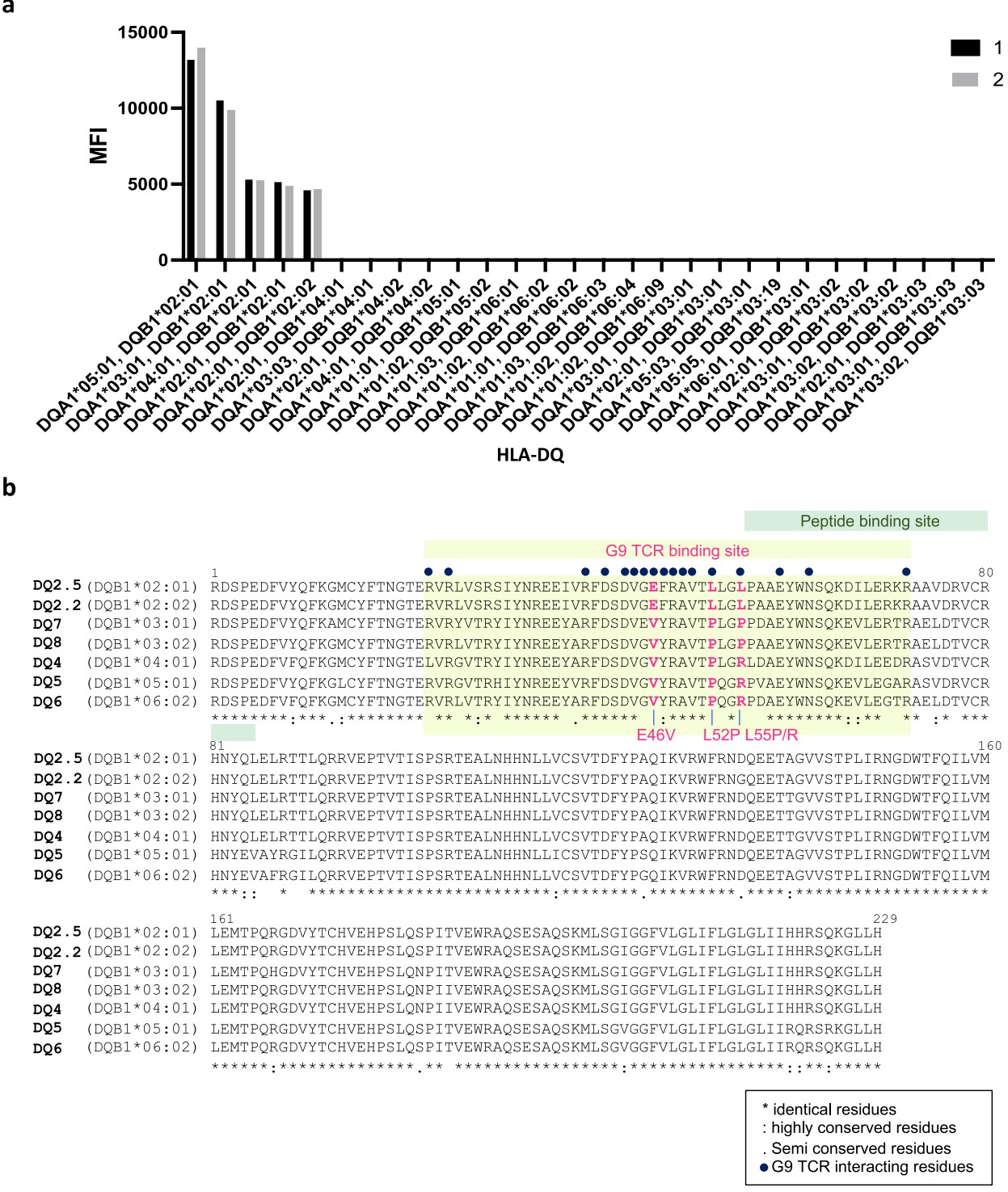

**Fig. 5 | G9 TCR tetramer display specific binding preference to HLA-DQ2, and multiple sequence alignment of HLA-DQ alleles. a** Detailed bar chart of reactivity of G9 TCR to HLA-DQ alleles, and the mean fluorescence signal (MFI) was measured as a read out ($n = 2$). **b** Multiple sequence alignment of HLA-DQB1 alleles. Residues are outlines with black asterisk (identical residues), colon symbol (highly conserved residues), period symbol (semi conserved residues), respectively. Residues involved in the G9 TCR-HLA-DQ2 interface are marked with dark blue dot and highlighted in light yellow. Peptide binding site is indicated as green highlight above the sequences. Residues in pink indicate the interacting residues that are distinct from other HLA-DQB1 alleles.

disruption of the arch-like architecture of HLA-CD4 by G9 TCR modality may explains the inhibition of subsequent T cell activation.

## Discussion

αβTCRs co-recognizing peptide antigens presented by antigen presenting molecules has been the central dogma of T cell mediated

responses over decades[4]. There are over 300 TCR-pMHC complex structures published to date[17], the vast majority of which adopt the canonical docking polarity. There are a few exceptions showing a reverse docking polarity[8,9,18]. Nevertheless, both docking modes involve specific interactions between TCR and peptide-MHC molecules, forming an 'end-to-end' co-recognition. Our findings uncover an

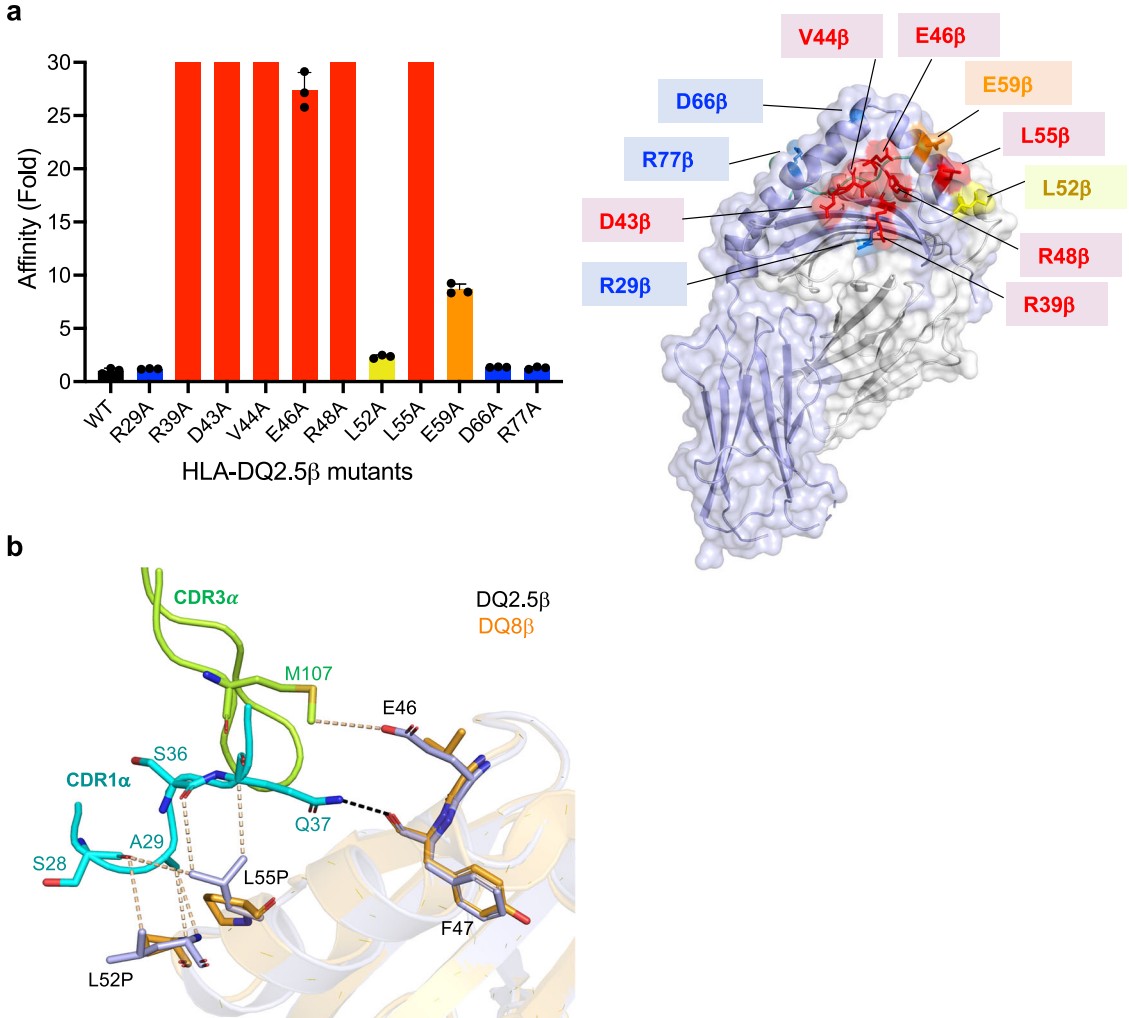

**Fig. 6 | Specific determinants in HLA-DQ2 define G9 TCR interaction. a** Left: Detailed bar chart of HLA-DQ2.5 mutations on G9 TCR binding affinity. The Y-axis represents the fold change in affinity of mutant HLA-DQ2.5 with respect to the wild-type HLA-DQ2.5 and the X-axis represents the position of HLA-DQ2.5 residues mutation. Three independent SPR experiments were performed and the error bars correspond to ± s.e.m. The impact of each mutation was classified as negligible (≤ 1-fold affinity decrease, blue), mild (1.5- 3-fold affinity decrease, yellow), moderate (3-fold to 10-fold affinity reduction, orange), or critical (>10-fold affinity decrease or no binding, red) shown on graph and on surface of pHLA; right: the footprint of energetically important HLA-DQ2 contact residues were shown as sticks and coloured according to the changed $K_D$ in fold as (**a**). **b** Superposed structure of HLA-DQ2.5 (light blue) and HLA-DQ8 (orange) β-chain at the G9 TCR CDRα loops interface. The CDR loops 1α, and 3α are highlighted in cyan and light green colour, respectively. Interacting residues are shown in sticks. All amino acids are indicated in single-letter abbreviations.

unexpected binding mode of the αβTCR-pMHC, whereby the TRAV12-1[+]-TRBV5-1[+]G9 TCR is oriented in reversed polarity but positioned over the side of the HLA-DQ2 β-chain, away from peptide binding cleft, forming an 'end-to-side' recognition. Accordingly, such recognition is markedly distinct from all available αβTCR-pMHC structures[1]. In addition to the lack of interaction with the HLA-DQ α-chain, G9 TCR bound independently of a broad spectrum of peptide antigens, regardless of peptide sequence identity, demonstrating an antibody-like binding characteristic for pan-HLA-DQ2. This discovery parallels the emerging paradigm of docking topologies that have been identified in γδTCR-ligand-MHC-like structures, i.e., CD1a and MR1, whereby the γδTCRs bind to the side of CD1 or MR1, forming an end-to-side recognition[14,15]. Moreover, a recent report of direct recognition of αβTCR and an intact foreign protein, R-phycoerythrin (PE), independently of MHC, further showcases variable recognition mode of an αβTCR[19].

The emergence of this atypical recognition mode of G9 TCR-MHC has raised a key question: what drives this docking topology and what was the G9 TCR selected on? Out of all HLA-DQ2.5 specific-CD4[+]T cell

clones sorted from the peripheral blood of an HLA-DQ2.5 homozygous coeliac disease donor, we only found one αβTCR bearing *TRAV12-1[+]/ TRBV5-1[+]* gene usage, which was reactive similarly to all HLA-DQ2 allotypes in a peptide-independent manner. While there are no other TRAV12-1[+]/TRBV5-1[+] TCRs in the PDB, the TCR-pMHC complexes utilizing either *TRAV12-1[+]* TCR (PDB ID: 3RGV[20], 6VRM[21], 7PBE[22]) or *TRBV5-1[+]* TCR (PDB ID: 5BRZ[23], 6XC9[12], 4P4K[24], 1ZGL[25]) gene usage adopted the consensus canonical docking mode. The G9 TCR does not appear to have any specific attributes to enable peptide-independent HLA binding, as the affinity of the interaction, and characteristics (i.e., BSA) at the interface falls within the typical range of TCR-pMHC interactions[1]. The ability of the G9 TCR to bind HLA directly echoes recent structural observations of non-HLA protein interactions with TCRs, including αβTCRs recognizing CD1 directly without co-contacting antigen[26–28], αβTCR recognizing intact foreign protein, R-phycoerythrin[19], and γδTCRs-MR1/CD1 binding in atypical binding modes[14,15,29]. Furthermore, in the context of thymic selection, we were unable to detect signalling in cell lines expressing the G9 TCR, the signalling threshold for thymic selection is low compared to peripheral

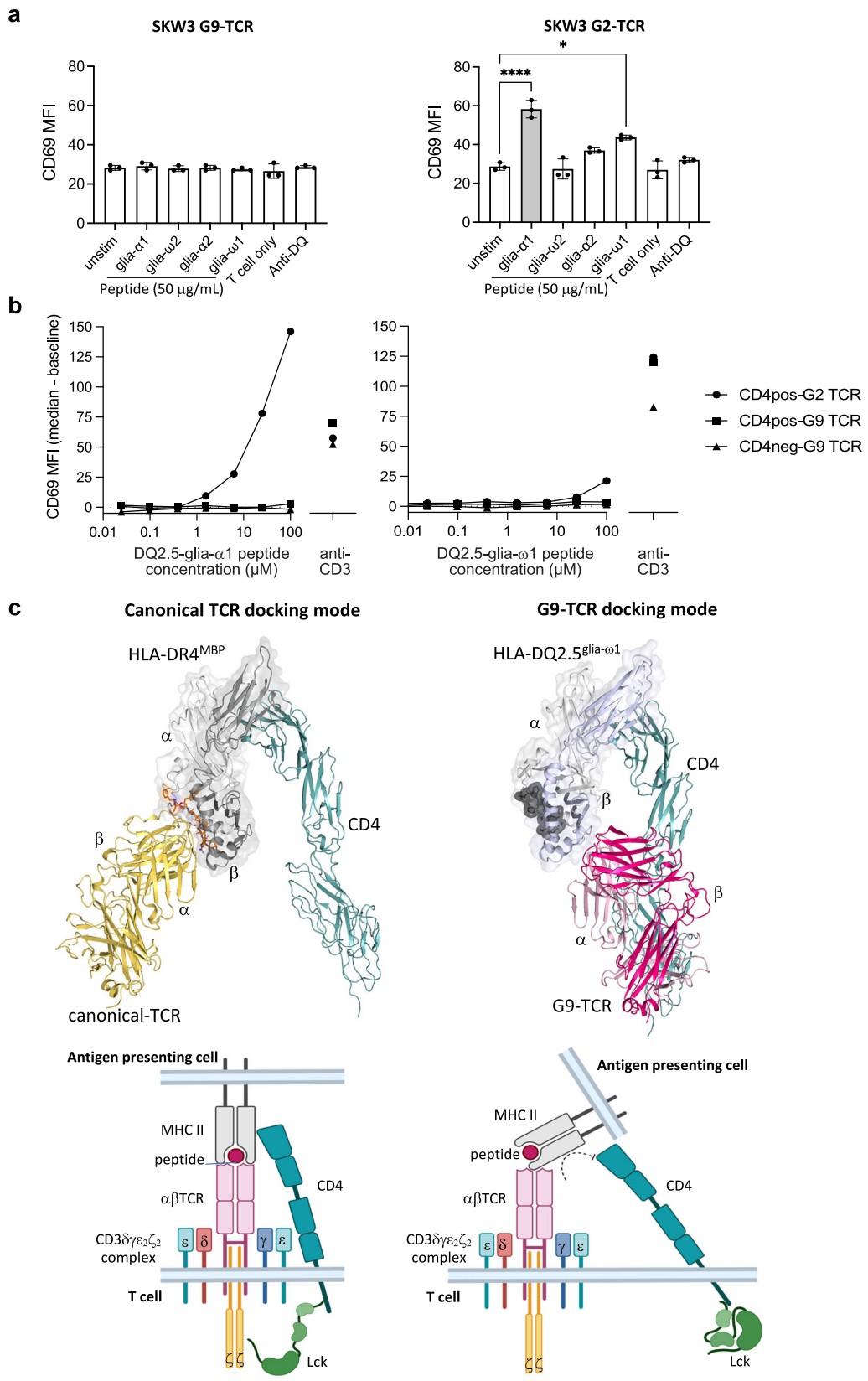

activation[30,31], and a weak signal transduced by G9 TCR recognition of MHC may be sufficient for development. Nevertheless, we also are mindful of the possibility that a different TCR docking mode that invokes co-recognition might occur during selection.

Moreover, the interactions of αβTCR-MHC governed by "energetic hotspots" showed that a Leucine residue at position 55 on the

HLA-DQ2 was responsible for critical contact with G9 TCR. This feature distinguished HLA-DQ2 from other HLA-DQ homologues, despite showing ~90% homology with HLA-DQ8 and HLA-DQ6. The inability to cross-react to the other 180 individual HLA class I and class II molecules, has affirmed the specificity of Leu[55] in mediating the interaction with the G9 TCR. However, based on intrinsic plasticity of TCRs, we

**Fig. 7 | G9 TCR-HLA-DQ2 interaction impedes T cell activation. a** Expression of CD69 on the surface of G9 TCR (left) and G2-TCR (right) transduced SKW3 cell line stimulated overnight with HLA-DQ2.5⁺ RAJI cells coated with 50 μg glia-α1, -α2, -ω1, -ω2 peptides r no peptide as control (denoted as unstim). Data is presented as the median mean fluorescence intensity (MFI) of CD69 ($n = 3$) and the error bars correspond to ± s.e.m. $P$-values were determined by one-way ANOVA with Dunnett's multiple comparison testing, $*P \leq 0.05$ and $****P \leq 0.0001$. **b** Expression of CD69 on the surface of G9 TCR transduced CD4⁺ and CD4⁻ Jurkat cell lines stimulated overnight with HLA-DQ2.5⁺ RAJI cells coated with serial dilution of 100-0.024 μM glia-α1 (left panel) or glia-ω1 (right panel) peptides, or 1μg anti-CD3. Data is presented as the median MFI at each peptide concentration less the baseline value (no peptide). Anti-CD3 antibody used as control. G2-TCR is a cross-reactive control TCR to HLA-DQ2.5$^{\text{glia-α1/-ω1}}$. **c** Top: overview structure of canonical TCR docking of MS2-3C8-TCR-HLA-DR4$^{\text{MBP}}$-CD4 complex (left; PDB: 3T0E), and superposed model of G9 TCR-HLA-DQ2.5$^{\text{glia-ω1}}$-CD4 complex (right); bottom: schematic representation of canonical TCR docking of CD3-TCR-pHLA-CD4 complex (left) and model of CD3-G9 TCR-HLA-DQ2.5$^{\text{glia-ω1}}$-CD4 complex (right). Figures were created BioRender. Lim, J. (2025) https://BioRender.com/w50m058.

certainly think that there is potentially other TCRs that can bind in a peptide-independent manner. Indeed, a number of TCRs have been described that bind to non-HLA-proteins[19,32].

The discovery of the new binding modality raised the question of its impact on TCR mediated signalling. To address the key physiological function of this natural TRAV12-1⁺/TRBV5-1⁺ G9 TCR, we stably expressed the TCR into T cell lines. We observed that G9 TCR ligation in this unconventional orientation was incapable of stimulating CD69 upregulation on cell lines, despite its high affinity. The result suggests that affinity alone is not a sufficient metric for the likelihood of eliciting a T cell signal, as exemplified by γδTCRs in which many of these that bind with similar affinities do not signal, or the TCRs are co-receptor independent[33]. The result also coincided with our superposed model of G9 TCR-HLA-DQ2 and MS2-3C8-TCR-HLA-DR4$^{\text{MBP}}$-CD4 complex revealing a steric clash with CD4 binding upon G9 TCR and HLA-DQ2 ligation. It is possible that CD4-MHC II binding is required to stabilize the G9 TCR-MHC II interaction. Although the affinity of the CD4-MHC II interaction is low[34], it is reported to increase substantially once MHC II has bound TCR[35]. As CD4 is still present but did not engage with G9 TCR-MHC II, it is thus still sequestering Lck. Although TCRs may not necessarily need CD4 to signal, it is different to having CD4 present but actively excluded from the interaction, because that will remove Lck as well.

Alternatively, CD4 binding of MHC II may be critical for delivery of Lck to CD3 to initiate signalling. Our previous study showed that signalling by a reversed docking MHC I-restricted TCR was prevented due to coreceptor mislocalisation of Lck relative to CD3 and was restored by liberating Lck from the coreceptor. In contrast, removal of CD4 (and thus liberation of Lck) here was not sufficient to restore signalling by the G9 TCR. It is well established that Lck preferentially binds to CD4 over CD8[36]. Thus, it is plausible that CD4 T cell activation is much more reliant on the coreceptor-directed delivery of Lck than CD8 T cells. Certainly, recent studies in which mice expressed mutated Lck that was unable to bind coreceptors showed a marked reduction in development and activation of CD4, but not CD8 T cells[37]. Apart from CD69, an early activation marker studied in this work, future studies could look at transducing into primary T cells and other activation markers (i.e., pERK). In summary, our findings reshape the range of known αβTCR binding modalities compatible with MHC binding.

## Methods

### Peptides
Peptides used for HLA-DQ2 were glia-ω1 (QPFPQPEQPFP), glia-ω2 (PQPEQPFPWQ), glia-α1 (QPFPQPELPYP), glia-α2 (FPQPELPYPQP), glutL1 (PASEQEQPV), and for HLA-DQ8 was glia-α1 (PSGEGSFQP-SENPQ). The peptides were synthesized by GL Biochem (China) and the integrity of the peptides was verified by reverse-phase HPLC and mass spectrometry.

### Tetramer-based magnetic enrichment and analysis of epitope specific CD4⁺ T cells in humans
Human experimental work was conducted according to the Australian National Health and Medical Research Council (NHMRC) Code of Practice. Patients with coeliac disease were recruited after provision of informed consent (Human Research Ethics Committees: Royal

Melbourne Hospital ID: 2020.162; The Walter and Eliza Hall Institute of Medical Research ID: 03/04). Tetramer-based magnetic enrichment was used for identification of epitope-specific CD4⁺ T cells in PBMC isolated from a patient with *DQA1\*05:01/DQB1\*02:01*⁺ coeliac disease (donor 0648, female on a gluten-free diet for ≥6 months). Peripheral blood was collected into lithium heparin vacutainers (Becton Dickinson) six days after the donor undertook a 3-day gluten challenge by consuming four slices of commercial white bread daily (approximately 10 g/day of wheat gluten)[38]. Mononuclear cells were obtained by centrifugation over Ficoll-Paque (GE Healthcare) and cryopreserved. Following thawing, PBMC were rested overnight in complete RPMI (cRPMI) containing RPMI1640 (Invitrogen, #21870), 10% foetal bovine serum (FBS; ThermoFisher Scientific, #26140079), Hepes (ThermoFisher Scientific, #15630), L-glutamine (ThermoFisher Scientific, #25030), sodium pyruvate (ThermoFisher Scientific, #11360), MEM Non-Essential Amino Acids (ThermoFisher Scientific, #11140), 2-mercaptoethanol (ThermoFisher Scientific, #21985023) and penicillin-streptomycin (ThermoFisher Scientific, #15140) at 37 °C and 5% $CO_2$. Cells were counted and 13.6 million PBMC were resuspended in 60 μl/1 ×10⁷ cells of sorter buffer [Phosphate Buffered Saline (PBS)/0.5% BSA/2 mM EDTA] and 20 μl/1 ×10⁷ cells of anti-human FcR blocking reagent (Miltenyi Biotec, #130-059-901) plus 8 μl/1 ×10⁷ cells of 500 nM dasatinib (Cell Signalling Technology, #9052S; final concentration 50 nM) and incubated for 30 min at 37 °C, before the addition of HLA-DQ2.5-glia-ω1-PE and HLA-DQ2.5-glia-ω2-APC tetramers (each at 10 μg/ml final concentration) for 1 h at room temperature. Cells were washed once with cold sorter buffer, then resuspended in 400 μl buffer plus 50 μl of each of anti-PE- and anti-APC- conjugated magnetic microbeads (Miltenyi Biotec, #130-048-801 and 130-090-855) and incubated for 30 min at 4 °C. Cells were washed twice in sorter buffer, resuspended in 3 ml buffer, and passed over a magnetic LS column (Miltenyi Biotec, #130-042-401) according to manufacturer's instructions. The initial flow-through was passed over the column twice, followed by 3 ×3 ml washes with buffer. The column was then removed from the magnet and bound cells eluted by pushing 5 ml of sorter buffer through the column. The eluted cells were then incubated for 30 minutes at 4 °C with a cocktail of conjugated antibodies to identify epitope-specific cells from total CD4⁺ T cell populations including CD14 (clone M5E2, BD Biosciences, Cat # 557923), CD19 (clone: HIB19, BD Biosciences, Cat # 557921), CD3 (clone UCHT1, Biolegend Cat. 300412), CD4 (clone SK3, BD Biosciences, Cat # 563550) and viability stain FVS700 (BD Horizon, Cat # BD564997) (Supplementary Table 7). The entire sample (including two rinses of sample tubes) were acquired on a FACSAria III cell sorter with FACSDiva 8.0.1 software (BD Immunocytometry Systems) following the gating strategy in Supplementary Fig. 1A. Live CD19⁻CD14⁻ CD3⁺ CD4⁺ HLA-DQ2.5$^{\text{glia-ω1}}$ and/or HLA-DQ2.5$^{\text{glia-ω2}}$ tetramer binding cells were single-cell index sorted into 96-well polymerase chain reaction (PCR) plates (Eppendorf #00301286) and stored at -80 °C until use.

### Analysis of epitope-specific T cell repertoires
For plates containing sorted HLA-DQ2.5$^{\text{glia-ω1}}$ and HLA-DQ2.5$^{\text{glia-ω2}}$ specific CD4⁺ T cells, mRNA was reverse-transcribed in 2.5 μl using the SuperScript™ VILO™ cDNA Synthesis Kit (ThermoFisher Scientific, # 11754) (containing 1x VILO™ reaction mix, 1x SuperScript™ enzyme

mix, 0.1% Triton X-100), and incubated at 25 °C for 10 min, 42 °C for 120 min and 85 °C for 5 min. The entire volume was then used in a 25 μl first-round PCR reaction with 1.5 U Taq DNA polymerase (Qiagen, #20120), 1x PCR buffer, 1.5 mM MgCl$_2$, 0.25 mM dNTPs and a mix of 40 human TRAV external sense primers and a human TRAC external antisense primer, along with 28 human TRBV external sense primers and a human TRBC external antisense primer[39] (Supplementary Table 8, each at 5 pmol/μl), using standard PCR conditions. For the second-round nested PCR, a 2.5 μl aliquot of the first-round PCR product was used in separate TRAV- and TRBV-specific PCRs, using the same reaction mix described above; however, a set of 40 human TRAV internal sense primers and a human TRAC internal antisense primer, or a set of 28 human TRBV internal sense primers and a human TRBV internal antisense primer, were used[39] (Supplementary Table 8). Second-round PCR products were visualized on a gel and positive reactions were purified with ExoSAP-IT reagent (ThermoFisher Scientific, #78201). Purified products were used as template in sequencing reactions with human internal TRAC or TRBC antisense primers and sequenced on an ABI 3730 DNA Sequencer at the Monash Micromon Genomics Facility (Monash University), and *TRBV* and *TRAV* gene usage was determined using IMGT/V-QUEST[40]. Selected P2A-linked TCRαβ gene constructs were custom ordered from Genscript and cloned into pMIGII (RRID: Addgene_52107; a gift from D.A.A. Vignali) vector and sequenced to confirm the correct TCR sequence.

## In-vitro TCR expression

Human embryonic kidney (HEK) 293 T cells (ATCC, #CRL-3216) were maintained in complete DMEM (cDMEM) containing Dulbecco's modified Eagle's medium (Invitrogen, #11960), 10% foetal bovine serum (FBS), Hepes, L-glutamine, sodium pyruvate, MEM non-essential amino acids, 2-mercaptoethanol and penicillin-streptomycin in a humidified incubator at 37 °C and 10% CO$_2$. HEK293T cells were plated at $3.5 \times 10^5$ cells/well of a six-well plate in 3.5 ml of cDMEM. The following day, 4.2 μl of FuGene 6 HD (Promega, #E2691) was added to 171 μl of OptiMEM (Invitrogen, #31985) in an Eppendorf tube and incubated for 10 min at room temperature (RT). The FuGene:OptiMEM mixture was then added dropwise to 700 ng of pMIGII encoding an αβTCR sequence and 700 ng of pMIGII encoding CD3γδε and ζ subunits[41] and incubated for a further 15 min at RT. The FuGene-OptiMEM-DNA mixture was then added dropwise to a well(s) of 293 T cells in the six-well plate and swirled to mix gently before returning to the incubator. After 48 hours, the culture medium was aspirated, and cells were detached from the plate by repeated washing with fluorescence-activated cell sorting (FACS) buffer (PBS + 0.1% bovine serum albumin). Transfected HEK 293 T cells were labelled with HLA-DQ2/DQ8 tetramers for 1 hour at RT, followed by APC conjugated anti-human CD3 antibody (clone UCHT1, Biolegend, #300412) and Live/Dead Aqua Blue viability stain (ThermoFisher Scientific, # L34957) (Supplementary Table 7). Cells were analysed on a BD LSRFortessa™ X20 with FACSDiva software (BD Immunocytometry Systems). Collected data were analysed using Flowjo v10.9.0 (FlowJo).

## Protein expression and purification

αβTCR was designed with extracellular portion of human *TRBC* constant regions and the *TRAV/TRBV* variable with an engineered disulphide linkage in the constant domains essentially as previously described[42]. Briefly, TCR α- and β-chains were expressed separately in *Escherichia coli* BL21 (DE3). Inclusion bodies purified and refolded in buffer containing 100 mM Tris pH 8.0, 5 M Urea, 0.4 M L-Arginine, 2 mM EDTA, 0.2 mM PMSF, 0.5 mM oxidised glutathione and 5 mM reduced glutathione for 72 h, at 4 °C with rapid stirring. The samples were dialyzed extensively in 10 mM Tris pH8.0 and purified on a DEAE (Cytiva) anion exchange column, followed by size exclusion (Superdex 200, 16/600; Cytiva), hydrophobic interaction (Hi Trap SP HP; Cytiva) and anion exchange (HiTrap Q HP column; Cytiva) chromatography.

The extracellular domains of HLA-DQ2.5 (*HLA-DQA1*05:01* and *HLA-DQB1*02:01*), HLA-DQ2.2 (*HLA-DQA*02:01* and *HLA-DQB1*02:02*), and HLA-DQ8 (*HLA-DQA*03:01* and *HLA-DQB1*03:02*) were covalently linked with invariant chain CLIP or gliadin epitopes at the N-terminus of DQ2 β-chain, and an enterokinase cleavage site prior to fos and jun leucine-zippers domain, followed by a BirA biotinylation site and polyHistidine tag at the C-terminus region. The HLA-DQ proteins were expressed in baculovirus-insect cell expression system using High Five insect cells (Trichoplusia ni BTI-TN-5B1-4, Thermo Fisher Scientific)[43]. Post-72 hours of HLA-DQ2 baculovirus infection in High Five cells, the soluble HLA-DQ proteins were purified from the cell culture supernatant via concentration and buffer exchange (10 mM Tris pH8.0, 500 mM NaCl) using tangential flow filtration (TFF) on a Cogent M1 TFF system (Merck Millipore), followed by immobilized metal ion affinity (Nickel-Sepharose 6 Fast Flow; Cytiva), and size exclusion (Superdex 200, 16/600; Cytiva) chromatography. Purified monomeric peptide-HLA-DQ2 was biotinylated using biotin protein ligase (BirA) in buffer containing 0.05 M bicine pH 8.3, 0.01 mM ATP, 0.01 mM MgOAc, 50 μM d-biotin, and 2.5 μg BirA. BirA was made according to protocols outlined in O'Callaghan C et al.[44].

## G9 TCR-transduced T cell lines

The G9 TCR α- and β-chains were cloned in the lentiviral vectors pLV-EF1α-MCS-IRES (Biosettia, #cDNA-pLV05, #cDNA-pLV06), and G9 TCR lentivirus was produced through co-transfection of these vectors along with viral packaging plasmids (pMD2.G, pMDLg/pRRE, pRSV-REV; Addgene) in HEK293T cells following the manufacturer's (Biosettia) protocol. The G9 TCR lentivirus was transduced into the SKW3 (DSMZ, ACC53) T-cell line for stable expression using the lentiviral transduction system.

TCR$^{null}$ CD4$^{null}$ Jurkat cells (TCRαβ- CD4- CD8-) were generated from TCR$^{null}$ Jurkat cells (TCRαβ- CD4+ CD8-) via CRISPR editing according to the Synthego Protocol "CRISPR Editing of Immortalized Cell Lines with RNPs Using Nucleofection". A synthetic guide RNA (sgRNA) for Exon 2 of the human CD4 gene (AGTGCCTAAAAGG-GACTCCC) was designed using the UCSC Genomics Institute Genome Browser (https://genome.ucsc.edu/cgi-bin/hgGateway)) and generated by Synthego. Briefly, 3.6 μl of 30 pmol/μl sgRNA and 0.6 μl Alt-R® S.p. Cas9 Nuclease V3 (Integrated DNA Technologies, Cat. 1081058) were complexed at a ratio of 9:1 sgRNA to Cas9 in a total volume of 15 μl with Nucleofector Solution™ + Supplement from the P3 Primary Cell 4D-Nucleofector™ X Kit (Lonza, Cat. V4XP-3032) for 10 minutes at RT before adding to $2 \times 10^5$ Jurkat cells in 5 μl Nucleofector™ solution. The cell-RNP solution was immediately transferred to a Nucleocuvette™ strip and cells were transfected in a 4D-Nucleofector™ X Unit (Lonza) using programme CL-120 for Jurkat cells. Cells were resuspended and transferred into a well of a 24-well plate containing 1 ml cRPMI with medium replacement after 24 hours. After 72 hours, cells were stained with anti-human CD3:APC, anti-human CD4:BUV395 and Live/Dead Aqua Blue viability stain (Supplementary Table 7) and live CD3-CD4- cells were individually sorted into wells of a 96-well tissue culture plate containing cRPMI. Clones were screened by flow cytometry for lack of CD4 expression and gDNA was extracted from selected clones using QuickExtract DNA Extraction Solution (Lucigen, Cat. QE9050). Genomic disruption of CD4 exon 2 was confirmed by standard PCR using primers designed 200–300 bp either side of the sgRNA site (forward 5′ CTCAGGTCCCTACTGGCTCA 3′; reverse 5′ CTACCCCATCCTC-CACCTTT 3′) and sequencing on an ABI 3730 DNA Sequencer at the Monash Micromon Genomics Facility (Monash University). Selected clone 2E7 had a 13-nucleotide deletion within human CD4 exon 2.

CD4$^+$ and CD4$^-$ Jurkat cells and HEK293T cells were maintained in cRPMI in a humidified incubator at 37 °C and 5% CO$_2$. HEK293T cells were plated at $1 \times 10^6$ cells/dish in a 15 cm tissue culture dish (Corning, #430599) in 10 ml of cRPMI. The following day, 30 μL of FuGene 6 HD (Promega, #E2691) was added to 470 μL of OptiMEM (Invitrogen,

#31985) in a microcentrifuge tube and incubated for 10 minutes at RT. The FuGene:OptiMEM mixture was then added dropwise to 4 µg of pMIGII encoding an αβTCR sequence, along with 4 µg of pPAM-E and 2 µg of pVSVg[45], and incubated for a further 15 minutes RT. The FuGene-OptiMEM-DNA mixture was then added dropwise to the HEK293T cell culture and swirled to mix gently before returning to the incubator. The next day, medium containing FuGene:OptiMEM:DNA was replaced with fresh cRPMI and incubated for 12 hours. Supernatant was removed approximately every 12 hours seven times and filtered through a 0.45 µm syringe driven filter. Polybrene (Sigma-Aldrich, #H9268) (6 µg/ml) was added to the supernatant before re-suspending CD4[+] or CD4[-] Jurkat cells in filtered retrovirus containing supernatant. After seven virus transfers, Jurkat cells were grown to confluency in fresh cRPMI.

## T cell stimulation assay

G9 TCR-transduced SKW3, CD4pos Jurkat and CD4neg Jurkat cells, G2 TCR-transduced SKW3 and CD4[+] Jurkat cells, and RAJI B cells (HLA-DQA*01:01, HLA-DQB1*05:01) were cultured in cRPMI at 37 °C in 5% CO₂. Synthetic peptides (glia-ω1, glia-ω2, glia-α1, and glia-α2) were added to $0.1 \times 10^6$ RAJI cells in wells of a 96 well cell culture plate and incubated for 1–2 h at 37 °C in 5% CO₂. Next, $0.1 \times 10^6$ SKW3 G9 TCR cells, SKW3 G2 TCR or non-transduced SKW-3 parental cells (TCR deficient; German Collection of Microorganisms and Cell Cultures; negative control) or $5 \times 10^4$ CD4[+]-G2 TCR, CD4[+]-G9 TCR or CD4[-]-G9 TCR Jurkat cells were added to RAJI cells or to a well coated with 1 µg anti-human CD3ε antibody (clone OKT3; Monash Antibody Discovery Platform, Monash University) and incubated overnight at 37 °C in 5% CO₂. The cells were then washed twice in FACS buffer (PBS containing 10% FCS) by centrifugation (350 g for 5 min), then stained with anti-human CD4 (clone SK3, BD Biosciences, Cat # 563550), and/or anti-human CD3 (clone UCHT1, Cat # 300412 or 563546, BD Biosciences) and Anti-Human CD69 (Clone FN50, Cat # 555533, BD Biosciences) (Supplementary Table 7) for 30–60 minutes on ice in the dark. The cells were washed twice with PBS by centrifugation followed by live/dead cell staining with Zombie NIR (Biolegend, Cat # 423106) for 30 min or with Live/Dead Fixable Aqua Blue (Invitrogen, ThermoFisher Scientific, Cat # L34957) for 10 min at 20 °C in the dark. After live/dead cells staining, the samples were washed 2-3 times in FACS buffer by centrifugation and subsequently analysed via flow cytometry (Fortessa X2c; BD Biosciences). Collected data were analysed using Flowjo v10.9.0 and plotted with GraphPad Prism. Three independent experiments were conducted.

## Crystallization, data collection and processing

The monomeric HLA-DQ2 proteins were treated with enterokinase to remove C-terminal tagging prior to complexing with G9 TCR for crystallization. G9 TCR-HLA-DQ2 ternary complexes were concentrated up to 8 mg/ml and undertook high throughput crystallisation screening at the Monash Molecular Crystallisation Platform (MMCP) using an automated robotic NT8 system. Crystal hits were further upscaled and optimised via hanging-drop vapour diffusion method in 24 well plates. Protein was mixed at 1:1 ratio with each crystal condition ("mother liquor") and equilibrated against 500 µl of mother liquor. The G9 TCR ternary complex crystals were obtained in conditions containing 8% Tacsimate pH 8.0, and 20–24% w/v PEG3350. Crystals were cryoprotected in the mother liquor well solution supplemented with 20% glycerol prior to flash freezing in liquid N₂. Diffraction data was collected on the MX2 Beamline of the Australian Synchrotron, using an Eiger x16M detector, subsequently auto processed and scaled with XDS and CCP4 Software Suite, version 8.0.

## Structure determination, refinement, and validation

G9 TCR-HLA-DQ2 crystal structures were solved by molecular replacement in PHASER (CCP4 Software Suite, version 8.0) using a separate search model for the αβTCR (PDB ID: 6V1A[46]) and HLA-DQ2.5[glia-ω1] (PDB ID: 6MFF[47]). Repeated rounds of model building in Coot (version 0.9.8.91)[48], manual and automated refinement using PhenixRefine (PHENIX[49], version 1.20.1-4487) were carried out. The quality of the ternary structures was validated at the Protein Data Bank (PDB) Data validation and deposition services website. αβTCR variable domain was numbered according to the IMGT unique numbering system[50]. Data processing and refinement statistics were summarized in Supplementary Table 1. Buried surface area (BSA) calculations were performed using programme Areaimol while contact analysis was performed using Contacts (CCP4 Software Suite, version 8.0). All structural figures were generated by PyMOL version 2.4.0.

## Surface Plasmon resonance

Affinity measurements were performed using SPR on a Biacore T200 instrument (Cytiva). Biotinylated HLA-DQ2 molecules was immobilized on a streptavidin (SA) sensor chip (Cytiva) with approximately 3000 response units. Biotinylated HLA-DQ8[glia-α1] was used as a reference cell. Serial dilutions of TCR were passed over the flow cell surface in 20 mM HEPES pH 7.5, 150 mM NaCl, 1 mM EDTA, and 0.005% TWEEN 20 at a flow rate of 10 µl/min. GraphPad Prism v.9.0 (GraphPad Software) was used for data analysis of sensorgrams from which curves were plotted. At least two independent experiments ($n \geq 2$) were performed for each TCR sample. Equilibrium response curves were normalised against the calculated maximum response and the measurements then combined. Data are shown as mean ± standard error of the mean, s.e.m. RA2.7 TCR[51] and G2 TCR were used as negative and positive control, respectively.

## Luminex assay

G9 TCR was screened against a panel of 180 individual HLA-class I and class II molecules (LABScreen™ Single Antigen HLA class I, LS2A01 and class II, LS1A04). R-Phycoerythrin (PE)-conjugated G9 TCR tetramer (5 µg per well) was incubated with microbeads coated with individual HLA-class I and class II molecules in 300 mM PBS (PBS-300) with 5% foetal calf serum (FCS; AusGeneX) for 30 min at room temperature in the dark. The microbeads were then washed three times with PBS-300 containing 0.05% Tween 20, centrifuged at 1300 g for 5 minutes each, and final resuspended in PBS-300. Binding of G9 TCR and HLA molecules were analysed on Luminex® FLEXMAP 3D® flow analyse through identification of the individual HLA allotype via unique microbead labelling and detection of tetramer fluorescence intensity on each microbead set. Normalized fluorescence values were calculated using HLA Fusion™ software suite (One Lambda) by subtracting background values using the following formula:

$$(S\#N - SNC\ bead) - (BG\#N - BGNC\ bead)$$

Where $S\#N$ is the sample-specific fluorescence value (trimmed mean) for bead $\#N$, $SNC$ is the sample-specific fluorescence value for the negative control (nude) bead, $BG\#N$ is the background negative control fluorescence value for bead $\#N$, and $BGNC$ bead is the background negative control fluorescence value for negative control bead. Binding levels of each HLA molecules were obtained by subtracting the mean fluorescence intensity (MFI) values of raw value from G9 TCR experiment with an isotype control (a PE-conjugate IgG). Two independent experiments were performed.

## Reporting summary

Further information on research design is available in the Nature Portfolio Reporting Summary linked to this article.

# Data availability

The X-ray crystal structures were deposited in the Protein Data Bank (PDB) with the following accession codes: G9 TCR-HLA-DQ2.5[glia-w1], 9EJG; G9 TCR-HLA-DQ2.5[CLIP], 9EJH; G9 TCR-HLA-DQ2.2[glutL1], 9EJI. All

data generated in this study are provided in the Supplementary Information or available from the authors. The raw numbers for charts and graphs are available in the Source Data File. Source data are provided with this paper.

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

## Acknowledgements
We thank the staff at the Australian Synchrotron for assistance with data collection, the staff at the Monash Macromolecular crystallization facility and the staff at Monash FlowCore. We thank Nathan Felix and Murray McKinnon for useful discussions.

## Author contributions
J.J.L. and C.M.J. performed the research, analysed the data and wrote the paper alongside J.R. T.J.L., H.T.D., M.T.T. conducted research. J.T.D. provided key reagents. J.R. and N.L.G. co-supervised the research, project administration. JR, funding acquisition. All authors reviewed and edited the paper.

## Competing interests
This work has been supported, in part, by Janssen Research & Development. The authors declare no competing interests.
