## [Transparent Peer Review file · Nature Communications]

A naturally selected $\alpha\beta$ T cell receptor binds HLA-DQ2 molecules without co-contacting the presented peptide

Corresponding Author: Professor Jamie Rossjohn

Version 0:

Reviewer comments:

Reviewer #1

(Remarks to the Author)

The results presented by Lim et al are novel and very interesting. They provide a different and complementary understanding of TCR binding and specificity to HLA/peptide complexes. The quality of the structural and biophysical studies is high, and supports well the conclusions and claims. Altogether, the importance, novelty and quality of the presented results make them worthy of publication in Nature Communications.

1) The authors should consider and discuss how the qualities/sequence identity of the presented peptides (used within this study) may have an effect of the appropriate positioning of the section of the alpha chain that seems to play a key role for adequate recognition by the TCR G9. Any allosteric effect? Overall, what is the role of the peptide in this case?

3) Although well written, I found that significant sections of the discussion part are slightly repetitive of what is already presented within the results section. In my opinion, the discussion should be a platform for a certain amount of reasoned speculation, and I therefore think that it would be very giving for immunologists and structural biologists alike, if the authors would expand much more on the implications of their discovery on 1) positive/negative selection, 2) cross-reactivity and 3) alloreactivity.

Positive-negative selection; What are the implications? How does it work? Can they speculate on how TCRs are potentially sterically hindered in their selection phases?

Implications for cross-reactivity even though the authors have demonstrated that the TCR is highly specific to DQ2? Could other TCRs that potentially bind to other HLA/peptide complexes in a similar way (not necessarily on L155) be cross-reactive? Can the authors speculate more on this?

Can the authors discuss and speculate on the importance and possibly the potential quantities of such TCRs as well as TCRs that are specific to proteins without any involvement of HLA presentation? In their opinion, how do all these very recent discoveries fit with common knowledge?

Minor comments:

- 1) Please introduce a table (supplementary material) in which the names and exact sequences of the peptides are given. It would be very good if the authors would indicate the peptide residues that are important for binding. As well as the length of each peptide. Maybe indicate the part of the peptide that is closest to the TCR G9?
- 2) There seems to be an error in the listing of the peptides in lines 90-92.
- 3) The quality of the resolution of the annotations (names of each used tetramer) should be improved in Figure S1b.
- 4) Peptide should be mentioned in Figure 4a.
- 5) Please remove the 'HLA' in the annotation for the crystal structure of the TCR/Db/NP complex in Figure 4b.
- 6) The authors state that no contacts were made with the HLA-DQ2 helical-chain and the glia-w1 peptide, with the closest distance between CDR loops and peptide being 17Å (Fig. 2C). Can the authors be more specific about how they exactly measured this distance more exactly? Does the measurement 'go through' the helical-chain or did the authors measure this distance using other considerations?

Adnane Achour

Reviewer #2

(Remarks to the Author)

Lim et al provide a thorough analysis of an interesting abTCR called "G9", identified by tetramer staining peripheral blood of an individual with celiac disease, and which binds to HLA-DQ2.5 and related HLA-DQ alleles in a peptide-independent manner. The peptide-independent, MHC-specific binding activity of the G9 TCR was confirmed by studies with TCR-transfected HEK293T cells and biophysical studies with purified recombinant TCR and MHC-peptide complexes, which revealed relatively tight-binding $K_d \sim 7\text{-}15 \mu\text{M}$ for a variety of DQ2.5 and DQ2.2 peptides. X-crystallography of three TCR-pMHC complexes and extensive mutagenesis studies revealed a conserved binding site for TCR variable loops on the side of the HLA-DQ beta-domain alpha helical region and the underside of the adjacent beta strands, well away from the HLA-DQ bound peptide, thus providing a mechanistic basis for the lack of peptide sequence specificity. This highly non-canonical, gdTCR-like mode of TCR-pMHC interaction was not able to support activation of transduced Jurkat recombinants, likely due at least in part to interference with productive co-receptor CD4 interactions. The data are clearly presented with highly informative figures. Overall, these results highlight the highly adaptable nature of the TCR binding site.

However, this report does not in any way "break[s] the TCR-pMHC co-recognition paradigm" as stated in the abstract. TCR binding sites are generated by essentially random recombination of gene segments with addition or removal of nucleotides at the junctions, and like antibodies would be expected to be able to bind with high affinity to a huge number of potential ligands. Indeed, abTCR that recognize a variety of non-MHC ligands have been reported previously, including R-PE as noted in the text (ref 19), but also CD155, CD102, and other proteins as shown by Al Singer's group. So the discovery that a TCR can bind an MHC protein away from the binding site does not really break new ground. The G9-HLA-DQ2.5 binding interaction is not likely to impact either positive or negative selection, since it does not result in initiation of TCR signaling pathways, and would be ignored by the developing immune system. The authors do not consider the possibility that the G9 TCR was selected by another, conventional MHC-peptide ligand, and that the HLA-DQ2.5 binding activity is a coincidental side reaction. It perhaps would not be surprising for such a coincidental binding interaction to be discovered in a screen of millions of TCRs against a library of MHC tetramers. A search of VDJdb shows that a conventional, peptide-specific human TCR recognizing HLA -A2 bound to the influenza peptide GILGFVTL in fact has been reported with essentially identical TCR α as G9, having TRAV12-2, TRAJ8, and CDR3 MGFQKL (van de Sandt Nat Immunol 2023 PMID:37749325). No TCR β sequence was reported for this TCR, so it is not known if can bind to HLA-DQ2.5, but it certainly raises the possibility that the G9 TCR might be able to bind to a different peptide-MHC complex in a canonical, peptide-specific manner.

Two other issues should be addressed:

The authors should reconsider their use of the term "recognize" to characterize the peptide-agnostic interaction of G9 TCR with HLA-DQ2.5. While G9 TCR clearly binds to HLA-DQ2.5 with reasonable binding affinity, this interaction does not lead to T cell activation, as shown clearly Fig 7ab, and emphasized repeatedly in the text. Conventional understanding of recognition in the context of T cell biology and the MHC-TCR literature would include interactions that lead to T cell activation, but not non-productive interactions that do not activate TCR signaling. This is stated clearly by the authors themselves in the second sentence of the introduction: "abT cell receptors (TCRs), expressed on the surface of T cells, become activated upon recognition of the antigenic peptide presented by MHC molecules." (emphasis added). Perhaps "bind" would be a more appropriate term than "recognize" to characterize this behavior.

One of the mutations in Figure 3 appears to be colored incorrectly. It might make more sense to group CDR3b E111A mutation with CDR3b R109a and CDR3a M107A, in the "substantially reduced" but not "critical" group since their error bars appear to overlap, and color this mutation orange not red. That would be consistent with the description in the text." For the b-chain of the G9 TCR, alanine substitution of Phe57b (CDR2b, Arg66b (FWb, Arg109b and 162 Glu111b (CDR3b showed substantial impact on HLA-DQ2 recognition (> 5-fold reduced affinity)."

Reviewer #3

(Remarks to the Author)

Lim et al. describe the discovery and characterization of a naturally occurring patient-derived T cell receptor (TCR) that demonstrates a novel mechanism of recognition for the class II MHC protein HLA-DQ2. Unlike conventional TCRs, this receptor binds to HLA-DQ2 in a peptide-independent manner via a distinctive side-on docking geometry. Despite the lack of peptide involvement, the TCR exhibits selective binding to HLA-DQ2 and does not recognize other class II MHC molecules, underscoring a unique mechanism of MHC specificity. Through a combination of structural, cellular and biophysical analyses, the study identifies key HLA-DQ2-specific residues that underlie this specificity. Interestingly, while the TCR demonstrates high-affinity binding (for a TCR), it is functionally inactive. The findings are noteworthy, as they challenge the prevailing paradigm of TCR-peptide-MHC co-recognition and provide new insights into the diversity of TCR recognition mechanisms; especially in understanding of the relationship between different TCR-pMHC binding geometries/interactions and their functional outcomes.

The study is methodologically robust, employing a wide array of techniques. These include sequencing for TCR identification, tetramer-based and cellular assays for functional characterization, X-ray crystallography for structural elucidation, and SPR-based analyses for biophysical characterization and structure validation. The inclusion of extensive SPR-based mutational scanning and a comprehensive Luminex assay screening 180 individual HLA molecules adds significant depth to the investigation.

Although the manuscript reflects an impressive amount of important work the following issues should be addressed:

Major point for revision:

1. The hypothesis that CD4 is differentially placed and this leads to a lack of signaling is interesting. But the affinity is high enough that, according to most models and indeed most data, some signaling should be apparent even without CD4. Thus while the model for CD4 differential placement is fine, the authors need to talk about why it doesn't signal even in the absence of CD4. Given all the talk about TCR signaling mechanisms, this needs to be addressed.

Other points:

2. There are no electron density maps included in manuscript. The authors need to include unbiased composite omit maps for the interacting surfaces and important residues, etc. Also CC1/2 values should be added to the X-ray table.

3. Where leucine-zippers, BirA biotinylation sites and polyHistidine tags were removed from pHLA complexes prior to crystallization or analysis?.

4. Figure 2d-g: please provide a detailed legend for the bond colors and types.

5. Figure 1c: for the bottom row, please add error bars in the SPR binding curves as in the row above and the supp data.

6. Clearly indicate negative and positive controls used in the SPR experiments.

7. Supplementary Figure 2 caption: specify the method of superposition in supp figure 2: all atom/backbone of the full complex or part of the complex. This is also applicable to wherever a superposition is carried out.

8. Provide the actual RMSD values. Only a range of RMSD values is mentioned in line 109.

9. Please specify whether the detailed interactions in Fig 2d-e are for HLA-DQ2.5 presenting glia- ω 1 peptide. This is not mentioned anywhere. Provide a supplementary figure with detailed interactions for the other 2 complexes. Consider highlighting the differences in interactions between the three complexes in a new supplementary figure along with Table S3-S5.

10. Simple models the structures with critical mutations (e.g., L55A) would add more perspective to the structural analysis and could be presented as new supp figures (these could be used, for example, to further emphasize how the mutations disrupt the interactions and/or discuss the possibility that the mutations change the overall architecture of the loop leading to non-binding/reduced affinity).

11. Line 417-419, 476: Correct the DQA and DQB chain names.

12. Line 520: Provide a brief biotinylation protocol.

13. Label the schematic in Figure 7c.

14. Lines 66 and 84: Include missing peptide sequences (of CLIP, glutL1, glia- α 1 and glia- α 2)

15. Line 131: Include the types of interactions here too as done in other places in this paragraph.

16. Line 143: Provide rationale for the selection of mutation sites (Type of interaction/bond length/number of interactions per residue etc.?)

17. Line 15: Although joint 1st authors are indicated, only one author is marked with an *

18. The manuscript has many formatting inconsistencies and typographical errors. The latter might be caught in revision and copyediting, but the formatting ones may not: For example:

- Supplementary Figure 1: use ω and α instead of w and a
- Supplementary Figure 2 caption: capitalize CLIP
- Supplementary Figure 2 caption: Remove space between FW and α
- Supplementary Figure 3: Complete the figure title in bold
- Supplementary Figure 3 and 4: Use correct " μ " symbol with the KD values on binding curves
- Supplementary Figure 4: Figure title in bold: ..toward point mutations..
- Be consistent in using abbreviations such as FW α , FW β , VdW, PBS etc.
- Be consistent in using either the three letter- or one letter-code for amino acids in figures.
- Line 125: hydrogen bonds (H-bonds)
- Line 126: Van der Waals (VdW) interactions
- Line 133: VdW
- Figure 2b pie chart: FW: Capitalize W

- Line 212: a panel of eleven alanine mutations
- Line 620: mean fluorescence signal
- Line 298: CDR4-MHC II (add space before II)
- Line 344: 37 °C (add space before units)
- Line 345: Phosphate Buffered Saline (PBS)
- Line 399: use abbreviated PBS
- Line 410: capitalize "i" in inclusion bodies, add a space after pH
- Line 413: add a space after pH
- Line 475-476, 481: replace pos and neg with + and –
- Line 484: use abbreviated PBS
- Line 522: add a space after pH and 150
- Line 533: add a space after 300, use abbreviated PBS
- Line 551: should be glia- ω 1, not glia-w1
- Line 569 and 572: capitalize CLIP

Reviewer #1

We thank the reviewer for the positive appraisal of our manuscript.

Comments:

- 1) The authors should consider and discuss how the qualities/sequence identity of the presented peptides (used within this study) may have an effect of the appropriate positioning of the section of the alpha chain that seems to play a key role for adequate recognition by the TCR G9. Any allosteric effect? Overall, what is the role of the peptide in this case?

We now added a supplementary Table 1 to include the list of peptides and sequence identity used in this study. We also added a supplementary Figure 3b to include the superposed structure of the HLA-DQ2 binding cleft and peptides (glia-w1, CLIP and glutL1), which revealed a conserved register from the P1-P9 pockets, with no significant changes in the conformation of helical regions of the DQ2 α - and β -chains.

We state: (Results section, page 4)

“The HLA-DQ2 peptide binding cleft was rigid with very limited deviation in the helix region of DQ2 α - and β -chains (**Fig. S3b**). Despite relatively low sequence identity of glia- ω 1 and CLIP (27%) or glutL1 (36%) peptides, the binding register of the peptide bound to HLA-DQ2 was conserved (**Fig. S3b, and Table S1**).”

We state: (Discussion section, page 9)

“In addition to the lack of interaction with the HLA-DQ α -chain, G9 TCR bound independently of a broad spectrum of peptide antigens, regardless of peptide sequence identity, demonstrating...”

Although well written, I found that significant sections of the discussion part are slightly repetitive of what is already presented within the results section. In my opinion, the discussion should be a platform for a certain amount of reasoned speculation, and I therefore think that it would be very giving for immunologists and structural biologists alike, if the authors would expand much more on the implications of their discovery on 1) positive/negative selection, 2) cross-reactivity and 3) alloreactivity.

We have shortened the discussion accordingly (Page 9 and 10), and have elaborated on implications for positive selection as follows:

“The G9 TCR does not appear to have any specific attributes to enable peptide-independent HLA binding, as the affinity of the interaction, and characteristics (i.e., BSA) at the interface falls within the typical range of TCR-pMHC interactions¹. The ability of the G9 TCR to bind HLA directly echoes recent structural observations of HLA-independent $\alpha\beta$ TCRs, $\alpha\beta$ TCRs recognizing CD1 directly without co-contacting antigen^{19, 26, 27, 28}, and $\gamma\delta$ TCRs binding in atypical binding modes^{14, 15, 29}”

“Furthermore, in the context of thymic selection, although we were unable to detect signaling in cell lines expressing the G9 TCR, the signaling threshold for thymic selection is low compared to peripheral activation, and a weak signal transduced by G9 TCR recognition of MHC may be sufficient for development^{30, 31}. Nevertheless, we also are mindful of the possibility that a different TCR docking mode that invokes co-recognition might occur during selection.”

“The inability to cross-react to the other 180 individual HLA class I and class II molecules, has affirmed the specificity of Leu⁵⁵ in mediating the interaction with the G9 TCR. However, based on intrinsic plasticity of TCRs, we certainly think that there is potentially other TCRs that can bind in a peptide-independent manner. Indeed, a number of TCRs have been described that bind to non-HLA proteins^{19, 30}.”

- 2) Positive-negative selection; What are the implications? How does it work? Can they speculate on how TCRs are potentially sterically hindered in their selection phases?

We now state: (Discussion section, page 9)

“Furthermore, in the context of thymic selection, although we were unable to detect signaling in cell lines expressing the G9 TCR, the signaling threshold for thymic selection is low compared to peripheral activation, and a weak signal transduced by G9 TCR recognition of MHC may be sufficient for development^{30,31}. Nevertheless, we also are mindful of the possibility that a different TCR docking mode that invokes co-recognition might occur during selection.”

- 3) Implications for cross-reactivity even though the authors have demonstrated that the TCR is highly specific to DQ2? Could other TCRs that potentially bind to other HLA/peptide complexes in a similar way (not necessarily on L155) be cross-reactive? Can the authors speculate more on this?

We state: (Discussion section, page 10)

“The inability to cross-react to the other 180 individual HLA class I and class II molecules, has affirmed the specificity of Leu⁵⁵ in mediating the interaction with the G9 TCR. However, based on intrinsic plasticity of TCRs, we certainly think that there is potentially other TCRs that can bind in a peptide-independent manner. Indeed, a number of TCRs have been described that bind to non-HLA proteins^{19,30}.”

- 4) Can the authors discuss and speculate on the importance and possibly the potential quantities of such TCRs as well as TCRs that are specific to proteins without any involvement of HLA presentation? In their opinion, how do all these very recent discoveries fit with common knowledge?

We state: (Discussion section, page 9)

“The G9 TCR does not appear to have any specific attributes to enable peptide-independent HLA binding, as the affinity of the interaction, and characteristics (i.e., BSA) at the interface falls within the typical range of TCR-pMHC interactions¹. The ability of the G9 TCR to bind HLA directly echoes recent structural observations of non-HLA protein interactions with TCRs, including $\alpha\beta$ TCRs recognizing CD1 directly without co-contacting antigen^{26, 27, 28}, $\alpha\beta$ TCR recognizing intact foreign protein, R-phycoerythrin¹⁹, and $\gamma\delta$ TCRs-MR1/CD1 binding in atypical binding modes^{14, 15, 29}.”

Minor comments:

- 1) Please introduce a table (supplementary material) in which the names and exact sequences of the peptides are given. It would be very good if the authors would indicate the peptide residues that are important for binding. As well as the length of each peptide. Maybe indicate the part of the peptide that is closest to the TCR G9?

We now added a **supplementary Table 1** for the peptide list and sequences used in this study and highlighted the peptide-binding register to HLA-DQ. To clarify, G9 TCR did not interact with peptide. Peptide residues are important for HLA-DQ binding.

We state (Page 3)

“To confirm specificity, we transiently expressed... as a control (Fig. 1b, Fig. S1b, Fig. S1c, and **Table S1**)”

2) There seems to be an error in the listing of the peptides in lines 90-92.

We initially sorted the T cells specific for DQ2.5^{glia- ω 1} or DQ2.5^{glia- ω 2} from peripheral blood. We then re-tested the TCRs (verified by re-expression and tetramer staining) using a panel of individual HLA-tetramers including DQ2.5^{glia- ω 1}, DQ2.5^{glia- ω 2}, DQ2.5^{glia-a1}, DQ2.5^{glia- α 2}, and DQ8^{glia- α 1}. Of interest, in one case, we found a TCR, namely clone B01, that cross-reactive for the highly homologous DQ2.5^{glia- ω 1} and DQ2.5^{glia- α 1} epitopes (**Fig. S1b**).

We state (Page 3)

“Instead, these TCRs were specific for DQ2.5^{glia- ω 1} or DQ2.5^{glia- ω 2}, or in one case, namely clone B01 cross-reactive for the highly homologous DQ2.5^{glia- ω 1} (PFPQPEQPF) and DQ2.5^{glia- α 1} (PFPQPELPY) epitopes (**Fig. S1b**).”

3) The quality of the resolution of the annotations (names of each used tetramer) should be improved in Figure S1b.

We have revised the **Fig. S1b** to a high-resolution figure.

4) Peptide should be mentioned in Figure 4a.

We now added the peptide name in **Fig.4a** and legend.

5) Please remove the ‘HLA’ in the annotation for the crystal structure of the TCR/Db/NP complex in Figure 4b.

We have removed it in the **Fig. 4**.

6) The authors state that no contacts were made with the HLA-DQ2 helical-chain and the *glia-w1* peptide, with the closest distance between CDR loops and peptide being 17Å (Fig. 2C). Can the authors be more specific about how they exactly measured this distance more exactly? Does the measurement ‘go through’ the helical-chain or did the authors measure this distance using other considerations?

As G9 TCR alpha chain was docked at the rear of the peptide binding pocket of HLA-DQ2, the distance between peptide and G9 TCR was directly measured through the helical chain of DQ2. This measurement simply explains that the CDR loops of the G9 TCR are positioned far from the peptide, unlikely to contribute in the TCR-MHC interaction.

We have revised the **Fig. 2c** to include the direct distance measurement of the CDR loop and C-terminal end of peptide.

Reviewer #2:

We thank the reviewer #2 for recognising our work as a thorough analysis of an interesting $\alpha\beta$ TCR and clearly presented with highly informative figures.

- 1) This report does not in any way “break[s] the TCR-pMHC co-recognition paradigm” as stated in the abstract.

To clarify, from the hundreds of TCR-pMHC structures solved, the TCR universally co-contacts the MHC and peptide. This is co-recognition. Our report details the TCR contacting the HLA only – and not the peptide. That is why our work breaks the co-recognition paradigm. Nevertheless, we have changed the abstract to state “deviates from” instead of “breaks”

- 2) The G9-HLA-DQ2.5 binding interaction is not likely to impact either positive or negative selection, since it does not result in initiation of TCR signaling pathways, and would be ignored by the developing immune system.

As we do not know what the selecting ligand is, or how this TCR engages such a ligand, it is difficult to speculate about signaling or selection in the thymus. However, this T cell exists in the periphery of a human, and so by necessity survives thymic selection. How this T cell achieves this is unclear. But we do now state in the discussion. (Page 9)

“..we also are mindful of the possibility that a different TCR docking mode that invokes co-recognition might occur during selection.”

- 3) The HLA-DQ2.5 binding activity is a coincidental side reaction

This cannot be stated with such conviction. This TCR was isolated from the periphery using MHC-tetramers, and we confirmed specificity via structural, biophysical and screening assays. We already know that other TCRs can bind non-MHC ligands, $\alpha\beta$ TCRs that are restricted to CD1 can break the co-recognition paradigm, and $\gamma\delta$ TCRs can bind MHC-like ligands in usual topologies. Our study here happens to represent the first example of an $\alpha\beta$ TCR not co-contacting the peptide. More studies are required to understand the extent of this observation in differing settings – and our work will certainly stimulate the field in this endeavour.

- 4) A search of VDJdb shows that a conventional, peptide-specific human TCR recognizing HLA -A2 bound to the influenza peptide GILGFVTL in fact has been reported with essentially identical TCRA as G9, having TRAV12-2, TRAJ8, and CDR3 MGFQKL (van de Sandt Nat Immunol 2023 PMID:37749325). No TCRb sequence was reported for this TCR, so it is not known if can bind to HLA-DQ2.5, but it certainly raises the possibility that the G9 TCR might be able to bind to a different peptide-MHC complex in a canonical, peptide-specific manner.

Thank you for raising this interesting point. Please note that the G9 TCR β -chain plays an important role in mediating contacts with the HLA-DQ2.5. As the TCR β -chain of the human TCR recognizing HLA-A2 bound to the influenza peptide remains unresolved, it is challenging to speculate whether the G9 TCR can recognise HLA-A2-influenza peptide – we did show however that it cannot bind HLA-A2 presenting an array of self-peptides.

- 5) The authors should reconsider their use of the term “recognize” to characterize the peptide-agnostic interaction of G9 TCR with HLA-DQ2.5.

We have changed recognise to bind.

- 6) One of the mutations in Figure 3 appears to be colored incorrectly.

We have fixed the color code in Fig.3.

Reviewer #3:

We thank the reviewer #3 for the positive appraisal and recognising our work as “ an impressive amount of important work” and “methodologically robust”.

Major comment:

- 1) The hypothesis that CD4 is differentially placed and this leads to a lack of signaling is interesting. But the affinity is high enough that, according to most models and indeed most data, some signaling should be apparent even without CD4. Thus while the model for CD4 differential placement is fine, the authors need to talk about why it doesn't signal even in the absence of CD4. Given all the talk about TCR signaling mechanisms, this needs to be addressed.

We state: (Discussion section, page 10 and 11)

“We observed that G9 TCR ligation in this unconventional orientation was incapable of stimulating CD69 upregulation on cell lines, despite its high affinity. The result suggests that affinity alone is not a sufficient metric for the likelihood of eliciting a T cell signal, as exemplified by $\gamma\delta$ TCRs in which many of these that bind with similar affinities do not signal, or the TCRs are co-receptor independent²⁶.”

“As CD4 is still present but did not engage with G9 TCR-MHC II, it is thus still sequestering Lck. Although TCRs may not necessarily need CD4 to signal, it is different to having CD4 present but actively excluded from the interaction, because that will remove Lck as well.”

“Apart from CD69, an early activation marker studied in this work, future studies could look at transducing into primary T cells and other activation markers (i.e., pERK).”

Other points:

- 2) There are no electron density maps included in manuscript. The authors need to include unbiased composite omit maps for the interacting surfaces and important residues, etc. Also CC1/2 values should be added to the X-ray table.

Yes. We have added a supplementary **Fig. S2**.

We now added CC1/2 values in the X-ray statistic **Table S2**.

- 3) Where leucine-zippers, BirA biotinylation sites and polyHistidine tags were removed from pHLA complexes prior to crystallization or analysis?

Yes. We have added the text at method section (Page 16):

“The monomeric HLA-DQ2 proteins were treated with enterokinase to remove C-terminal tagging prior to complexing with G9 TCR for crystallization.”

- 4) Figure 2d-g: please provide a detailed legend for the bond colors and types.

Thank you. We have added in the Figure legend 2 (Page 19):

“The H-bonds, VdW, and salt bridges were displayed as black, light beige, and red dash lines, respectively.”

- 5) Figure 1c: for the bottom row, please add error bars in the SPR binding curves as in the row above and the supp data.

The deviation in the bottom row panels were very subtle due to high response unit, thus the error bars were overlapped by the dot symbols.

- 6) Clearly indicate negative and positive controls used in the SPR experiments.

We state (Page 17, Method section)

“RA2.7 TCR⁴¹ and G2 TCR were used as negative and positive control, respectively.”

- 7) Supplementary Figure 2 caption: specify the method of superposition in supp figure 2: all atom/backbone of the full complex or part of the complex. This is also applicable to wherever a superposition is carried out.

The superposition of complexes was applied using Ca backbone of HLA-DQ2.

We state in the text: (Page 4)

“Alignment of three complexes on C α backbone of HLA-DQ2.5^{glia- ω 1} complex revealed...”

- 8) Provide the actual RMSD values. Only a range of RMSD values is mentioned in line 109.

We state in the text: (Page 4)

“...with a root mean square deviation (r.m.s.d) value of 0.13 Å and 0.35 Å for HLA-DQ2.5^{CLIP} complex, and DQ2.2^{glutL1} complex, respectively”

- 9) Please specify whether the detailed interactions in Fig 2d-e are for HLA-DQ2.5 presenting glia- ω 1 peptide. This is not mentioned anywhere. Provide a supplementary figure with detailed interactions for the other 2 complexes. Consider highlighting the differences in interactions between the three complexes in a new supplementary figure along with Table S3-S5.

We state in the **Fig. 2** legend: (Page 18)

“Overall G9 TCR CDR loops docking on HLA-DQ2.5^{glia- ω 1}..., **g** CDR3 β with HLA-DQ2.5^{glia- ω 1} are shown.”

We now added the detailed interactions of other two complexes in the **Fig S3c-f**. The differences in interactions were highlighted.

We state in the text: (Page 5)

“Detailed interactions of G9 TCR and HLA-DQ2.5^{CLIP} or DQ2.2^{glutL1}, respectively, are highly conserved, with very subtle deviations at HLA-DQ2 (Arg²³ and Asn⁶²) and G9 TCR (Gln¹¹⁰) (**Fig. S3c-f**).”

- 10) Simple models the structures with critical mutations (e.g., L55A) would add more perspective to the structural analysis and could be presented as new supp figures (these could be used, for example, to further emphasize how the mutations disrupt the interactions and/or discuss the possibility that the mutations change the overall architecture of the loop leading to non-binding/reduced affinity).

Thank you. We now added a supplementary **Fig. 7** corresponded to mutations at **Fig. 6a** and **6b** in a structural view to explain the impact of interactions with G9 TCR.

11) Line 417-419, 476: Correct the DQA and DQB chain names.

Thank you. We have revised the chain names.

We state (Page 14)

“The extracellular domains of HLA-DQ2.5 (*HLA-DQA*05:01* and *HLA-DQB1*02:01*), HLA-DQ2.2 (*HLA-DQA*02:01* and *HLA-DQB1*02:02*), and HLA-DQ8 (*HLA-DQA*03:01* and *HLA-DQB1*03:02*)...”

12) Line 520: Provide a brief biotinylation protocol.

We state in the manuscript: (Method section, Page 14)

“Purified monomeric peptide-HLA-DQ2 was biotinylated using biotin protein ligase (BirA) in buffer containing 50 mM bicine pH 8.3, 0.01 mM ATP, 0.01 mM MgOAc, 50 μ M d-biotin, and 2.5 μ g BirA. BirA was made according to protocols outlined in O'Callaghan C *et. al*⁶⁶”

13) Label the schematic in Figure7c.

Thank you. We have revised the labels.

14) Lines 66 and 84: Include missing peptide sequences (of CLIP, glutL1, glia- α 1 and glia- α 2)

Thank you. We have revised the peptide sequences accordingly.

15) Line 131: Include the types of interactions here too as done in other places in this paragraph.

We state in the text: (Page 5)

“...formed multiple contacts with Arg²³ β and Asp⁴³ β of HLA-DQ2.5 β -chain via salt bridges, H-bonds, and VdW interactions (**Fig. 2f and Table S4**).”

16) Line 143: Provide rationale for the selection of mutation sites (Type of interaction/bond length/number of interactions per residue etc.?)

Thanks.

We state in the text: (Page 5)

“Twelve residues of CDR loops that form multiple contacts (H-bonds, VdW and/or salt bridge) with HLA-DQ2, and three residues located at the crystallographic packing region or at the interface with no contact with HLA-DQ2 were selected.”

17) Line 15: Although joint 1st authors are indicated, only one author is marked with an *

Thanks. We have deleted the asterisk symbol and revised the text.

18) Supplementary Figure 1: use ω and α instead of w and a

Thanks. We have revised the symbols.

19) Supplementary Figure 2 caption: capitalize CLIP

Thanks. We have revised the text.

20) Supplementary Figure 2 caption: Remove space between FW and α

Thanks. We have revised the text.

21) Supplementary Figure 3: Complete the figure title in bold

Thanks. We have resolved the format.

22) Supplementary Figure 3 and 4: Use correct " μ " symbol with the KD values on binding curves

Thanks. We have corrected the unit.

23) Supplementary Figure 4: Figure title in bold: ..toward point mutations..

Thanks. We have revised the format.

24) Be consistent in using abbreviations such as FW α , FW β , VdW, PBS etc.

Thanks. We have revised the format.

25) Be consistent in using either the three letter- or one letter-code for amino acids in figures.

Thanks. We have revised the amino acids in figures in one letter code.

26) Line 125: hydrogen bonds (H-bonds)

Thanks. We have revised the format.

27) Line 126: Van der Waals (VdW) interactions

Thanks. We have revised the format.

28) Line 133: VdW

Thanks. We have revised the format.

29) Figure 2b pie chart: FW: Capitalize W

Thanks. We have revised the format.

30) Line 212: a panel of eleven alanine mutations

Thanks. We have revised the text.

31) Line 620: mean fluorescence signal

Thanks. We have revised the text.

32) Line 298: CDR4-MHC II (add space before II)

Thanks. We have revised the text.

33) Line 344: 37 °C (add space before units)

Thanks. We have revised the text.

34) Line 345: Phosphate Buffered Saline (PBS)

Thanks. We have revised the text.

35) Line 399: use abbreviated PBS

Thanks. We have revised the text.

36) Line 410: capitalize “i” in inclusion bodies, add a space after pH

Thanks. We have revised the text.

37) Line 413: add a space after pH

Thanks. We have revised the text.

38) Line 475-476, 481: replace pos and neg with + and –

Thanks. We have revised the text.

39) Line 484: use abbreviated PBS

Thanks. We have revised the text.

40) Line 522: add a space after pH and 150

Thanks. We have revised the text.

41) Line 533: add a space after 300, use abbreviated PBS

Thanks. We have revised the text.

42) Line 551: should be glia- ω 1, not glia-w1

Thanks. We have revised the text.

43) Line 569 and 572: capitalize CLIP

Thanks. We have revised the text.